# Engineering subtilisin proteases that specifically degrade active RAS

Yingwei Chen[1], Eric A. Toth [2,3,4], Biao Ruan[1], Eun Jung Choi[1], Richard Simmerman[1], Yihong Chen[2], Yanan He[2], Ruixue Wang[2], Raquel Godoy-Ruiz[2,4,5], Harlan King [2,6], Gregory Custer [2,7], D. Travis Gallagher[2,6], David A. Rozak[8], Melani Solomon[2,7], Silvia Muro[2,7,9], David J. Weber [2,3,4,5], John Orban[2,10 ✉], Thomas R. Fuerst [2,11 ✉] & Philip N. Bryan [1,2 ✉]

We describe the design, kinetic properties, and structures of engineered subtilisin proteases that degrade the active form of RAS by cleaving a conserved sequence in switch 2. RAS is a signaling protein that, when mutated, drives a third of human cancers. To generate high specificity for the RAS target sequence, the active site was modified to be dependent on a cofactor (imidazole or nitrite) and protease sub-sites were engineered to create a linkage between substrate and cofactor binding. Selective proteolysis of active RAS arises from a 2-step process wherein sub-site interactions promote productive binding of the cofactor, enabling cleavage. Proteases engineered in this way specifically cleave active RAS in vitro, deplete the level of RAS in a bacterial reporter system, and also degrade RAS in human cell culture. Although these proteases target active RAS, the underlying design principles are fundamental and will be adaptable to many target proteins.

[1] Potomac Affinity Proteins, North Potomac, MD, USA. [2] Institute for Bioscience and Biotechnology Research, Rockville, MD, USA. [3] Marlene and Stewart Greenebaum Cancer Center, University of Maryland School of Medicine, Baltimore, MD, USA. [4] Center for Biomolecular Therapeutics, Rockville, MD, USA. [5] Department of Biochemistry and Molecular Biology, University of Maryland School of Medicine, Baltimore, MD, USA. [6] National Institute of Standards and Technology and the University of Maryland, Rockville, MD, USA. [7] Department of Bioengineering, University of Maryland, College Park, MD, USA. [8] Unified Culture Collection, United States Army Research Institute of Infectious Diseases, Fort Detrick, MD, USA. [9] Institute for Bioengineering of Catalonia of the Barcelona Institute of Science and Technology & Institution of Catalonia for Research and Advanced Studies, Barcelona, Spain. [10] Department of Chemistry and Biochemistry, University of Maryland, College Park, MD, USA. [11] Department of Cell Biology and Molecular Genetics, University of Maryland, College Park, MD, USA. ✉email: jorban@umd.edu; tfuerst@umd.edu; pbryan@potomac-affinity-proteins.com

Engineering proteases that cleave specific signaling proteins in active conformations would open new possibilities to study, regulate, and reprogram signaling pathways[1]. Our target here was the active state of the rat sarcoma (RAS) oncoprotein. RAS is a small GTPase that responds to growth factors, activates downstream effector molecules (such as those in the MAPK pathway), and stimulates cell growth. Three RAS isoforms (HRAS, KRAS, and NRAS) are the primary regulators of cell signaling pathways. All three isoforms coexist in cells and have distinct but overlapping roles in signaling[2,3]. Inherent to the function of RAS is a switch between its inactive (GDP-bound) and active (GTP-bound) forms. Two regions, termed switch 1 (amino acids 30–38) and switch 2 (amino acids 59–76), undergo structural changes as RAS cycles between the GDP-bound and GTP-bound forms in all RAS isoforms. The GTP-bound conformation drives the cascade of signaling effects.

Based on these observations we have engineered the serine protease subtilisin to target the conserved QEEYSAM sequence in switch 2 of RAS (amino acids 61–67, Fig. 1A). Examination of several crystal structures of active RAS identified considerable conformational heterogeneity in switch 2 induced by a γ-phosphate on the nucleotide cofactor[4–6]. We hypothesized that these structural changes would uncover a cryptic cleavage site, making the QEEYSAM target sequence more vulnerable to proteolysis in active RAS than in the inactive form. Motifs of this type generally occur in amphipathic helices because most of the amino acids have high α-helical propensity and also because the spacing between the large hydrophobic amino acids (Y and M) matches a helical periodicity.

Natural subtilisins have broad sequence specificity, but there is a wealth of information about engineering mutations that alter specificity[7–14]. The major challenge to designing high-specificity proteases, however, is that specificity based on differential substrate binding falls far short of natural processing proteases. In many natural proteases the cognate sequence influences the chemical steps in peptide hydrolysis and not just binding steps[15]. To engineer highly sequence-specific subtilisins we combined two previous observations. First, mutations at remote binding pockets for substrate side chains (sub-sites) can distort the subtilisin active site[9,11]. Second, mutating a catalytic amino acid in an enzyme may allow chemical rescue with a small molecule cofactor[16,17]. We leveraged these observations to create an engineered protease that requires both conformational and chemical rescue by the substrate and cofactor, respectively, to achieve high levels of activity. In particular, for the fully engineered RAS-specific proteases described here, binding of the cognate QEEYSAM sequence results in structural changes that are transmitted from the binding pockets to the active site and enable cofactor

activation. In contrast, binding of non-cognate sequences adversely affects the active site and in fact antagonizes cofactor binding. These cofactor-dependent subtilisins strongly prefer dynamic regions of proteins (such as switch 2 in active RAS) over structured regions because efficient cleavage requires interactions with P5 to P1' amino acids in an extended conformation (Fig. 1b).

Engineered proteases were tested in vitro, in engineered E. coli, and in human cell culture. Major findings are the following: (1) Engineered proteases cut the QEEYSAM sequence in a synthetic peptide substrate with high specificity and are tightly controlled by the cognate cofactor; (2) X-ray crystal structures of protease-substrate-cofactor complexes reveal interactions involved in substrate recognition and cofactor activation; (3) The QEEYSAM sequence in native RAS is cut in response to the cognate cofactor; (4) RAS-specific proteases cut active (GTP) RAS 60-80 times faster than the inactive (GDP) form; (5) NMR analysis reveals that the QEEYSAM sequence is more dynamic in active RAS than the inactive form; (6) The level of RAS within E. coli cells can be regulated by co-expression of the RAS gene with different protease and cofactor combinations; (7) A RAS-specific protease can destroy RAS in mammalian cells.

## Results

**Protease engineering.** The engineering process to develop specificity for the RAS sequence *QEEYSAM* involved extensive modification of the *Bacillus* protease, subtilisin BPN'[16–24], a canonical serine protease in which the scissile peptide bond is attacked by a nucleophilic serine (S221). The nucleophilicity of S221 results from its interactions with the catalytic histidine (H64) and aspartic acid (D32) that together form a charge relay system[25]. Most substrate contacts are with the first five amino acids on the acyl side of the scissile bond (denoted P1 through P5)[26] and the first amino acid on the leaving group side (denoted P1', Fig. 1b). Corresponding sub-sites on subtilisin are denoted S1', S1, S2, etc. Natural subtilisins have a strong preference for a hydrophobic amino acid at S1 and S4 sub-sites (Fig. 1B), but little discrimination for a particular hydrophobic amino acid[27,28]. The key to engineering high sequence-specificity was linking interactions in amino acid sub-sites to the rate of the first chemical step (acylation). In our design strategy, we considered two well-documented properties of proteases. The first is that certain mutations at sub-sites alter the conformation of remote catalytic residues[9,11]. The second observation is that mutating a catalytic amino acid radically decreases activity of an enzyme[29,30] but in certain cases allows some chemical rescue of activity by an exogenous small molecule that mimics the mutated amino acid[31–33]. The design strategy was based on the hypothesis that a linkage

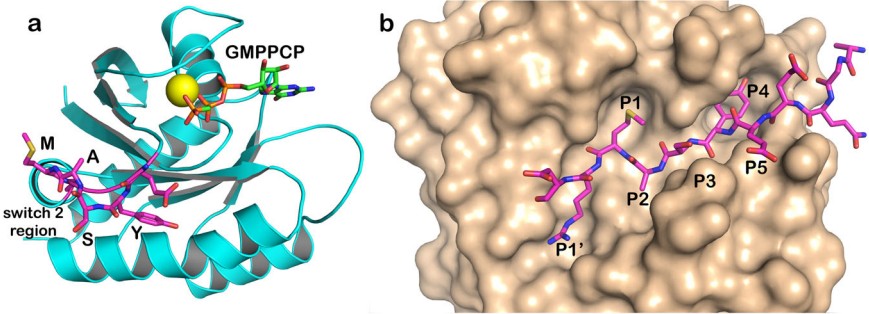

**Fig. 1 Target sequence (QEEYSAM) in RAS and in the active site of a RAS-specific protease. a** Structure of RAS with a bound GTP analog (PDB code 6Q21)[5,6], highlighting the YSAM site in Switch 2. **b** RAS-specific protease based on an X-ray structure of 3BGO.pdb[17]. Cognate sequence QEEYSAM-RD is modeled in the binding cleft. Substrate residues are denoted P1 through P5, numbering from the scissile bond toward the N-terminus of the substrate. The substrate amino acid on the leaving group side is denoted P1'.

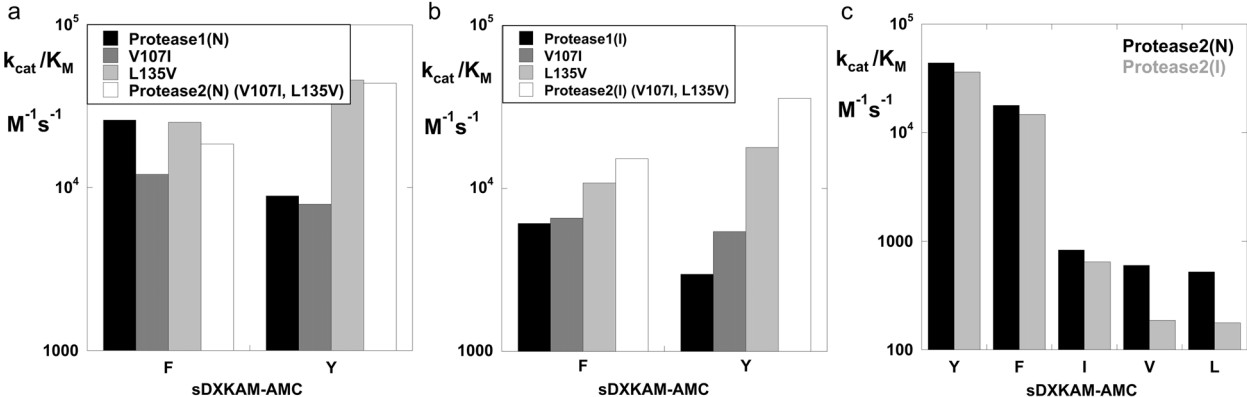

**Fig. 2 $k_{cat}/K_M$ as a function of mutation in the P4 pocket. a** Protease1(N) variants in 1 mM nitrite. **b** Protease1(I) variants in 10 mM imidazole. **c** Comparison of P4 specificity for Protease2(N) in 1 mM nitrite and Protease2(I) in 10 mM imidazole for the five highest activity sDXKAM-AMC substrates.

between substrate binding and chemical rescue can be created by mutating sub-sites to optimize interactions with the desired cognate sequence and combining these with mutation of a catalytic amino acid.

To begin testing this hypothesis robustly, we generated two different active site variations: D32G variants that are activated by nitrite or azide[16,17] and H64G variants that are activated by imidazole. Initial design was based on the X-ray structure of a D32A variant of subtilisin (**SBT189**, 3BGO.pdb)[17]. The structure shows the enzyme complexed with a cognate peptide (LYRAL) and azide bound in the position normally occupied by the catalytic D32. The substrate-binding pockets and the azide binding site form an interconnected network (Fig. S1)[8,11,34,35]. The theory is that binding at one sub-site can influence interactions in other parts of the network. The desired cognate sequence QEEYSAM-RD for RAS was modeled into the binding cleft of the 3BGO.pdb structure. Amino acid substitutions were introduced into the model and protein-protein and protein solvent interactions were evaluated by visual inspection[36]. Based on this analysis, as well as earlier engineering work, we designed mutations in the catalytic region, the S1 pocket, and the S4 pocket to create a nitrite-dependent (D32G) protease and an imidazole-dependent (H64G) protease (Table S1). These are denoted Protease1(N) and Protease1(I), respectively. These mutants were expressed, purified, and characterized for activity and specificity with a substrate series that was originally used to characterize **SBT189**: sDXKAM-AMC, where X = Y, F, I, or L[16,17,22,24]. AMC is the fluorogenic leaving group, 7-amino-4-methylcoumarin. Activity of both proteases is highest for P4 = F with lower activity for P4 = Y, I, and L (Fig. S2). The ability of other naturally occurring anions to activate Protease1(N) was examined using nitrite, formate, acetate, carbonate, and chloride (Fig. S3A). The ability of imidazole derivatives to activate the Protease1(I) was examined using imidazole, 4-hydroxymethyl-imidazole, imidazolepropionic acid, imidazoleacetic acid, and histamine (Fig. S3B). Consistent with earlier results[17], nitrite is the strongest activator of Protease1(N), with formate and chloride activating weakly at 10 mM concentration[17]. Imidazole is the strongest activator of Protease1(I) with 4-hydroxymethyl-imidazole) weakly activating at 10 mM concentration. As expected, none of the anions affect the activity of Protease1(I) and none of the imidazole compounds affect the activity of Protease1(N).

The next step was to determine the X-ray crystal structure of Protease1(N) in complex with a cognate peptide LFRAL (6UBE.pdb) and use this structure to model additional mutations in the S4 pocket. To increase specificity for P4 = Y, we introduced the single and double mutations of V107I and L135V into Protease1

(N) and Protease1(I) and examined the activity of the mutants on P4 = F vs. Y substrates. (Fig. 2a). The double mutants (denoted Protease2(N) and Protease2(I), Table S1) exhibited considerable preference for the substrate sD**Y**KAM-AMC. When P4 is F instead of Y, $k_{cat}/K_M$ falls by more than two-fold. All other variations at P4 result in a greater than 50-fold decrease in $k_{cat}/K_M$ (Fig. 2b, c). Further evaluation was carried out with peptide-AMC substrates with variations at P1 and P2. As previously observed for D32 mutants, H64G mutants also show high preference for P1 = M or L, and a moderate preference for P2 = A (Fig. S4)[16,17]. A thorough sub-site analysis previously performed on the natural subtilisins *BPN'* and *lentus* documents their high activity again a broad range of substrates sequences[37,38].

Finally, we examined variations to increase activity for P3 = S and P5 = E. Protease mutants at 101 (S, K, R) and 103 (Q, R) were evaluated using the peptide substrates QEXYSAM-AMC, where X = E, R, I, or L and QEEYXAM-AMC, where X = S, R, or E. The S101K mutants gave a high preference for the desired cognate sequence QEEYSAM. The S101K mutants were further analyzed for activity using the peptide substrate QEE**Y**SAM-AMC. Figure 3a compares $k_{cat}/K_M$ as a function of nitrite concentration for Protease2(N) and the S101K mutant. Figure 3b compares $k_{cat}/K_M$ as a function of imidazole concentration for Protease2(I) and the S101K mutant. The S101K mutation in both Protease2(N) and Protease2(I) increases the $k_{cat}/K_M$ by ~8-fold in 1 mM cofactor. The two S101K mutants are denoted as RAS-specific proteases RASProtease(N) and RASProtease(I). For reference, the activity of a previously engineered protease (**SBT160**) with a complete catalytic triad and preference for P4 = F or Y is also shown[24]. As expected, **SBT160** activity is unaffected by nitrite or imidazole (Fig. 3a, b).

**Synergy between conformational and chemical rescue**. Comparing nitrite activation in RASProtease(N) with the cognate substrate (QEE**Y**SAM-AMC) versus a near cognate (QEE**I**SAM-AMC) revealed a linkage between cognate substrate binding and cofactor activation (Fig. 4a). In 1 mM nitrite $k_{cat}/K_M$ is ~ 100-fold greater for P4 = Y versus I. The low activity of RASProtease(N) with the near cognate substrate is partly because of weaker substrate binding (5.5-fold less for P4 = I than P4 = Y) but mostly because of a lack of cofactor activation when P4 = I. A similar phenomenon is observed with RASProtease(I). The $k_{cat}/K_M$ value is ~300-fold higher for the cognate P4 = Y versus P4 = I in 1 mM imidazole (Fig. 4B).

**Structural analysis of protease interactions with the switch 2 target sequence**. To understand the structural basis for specificity

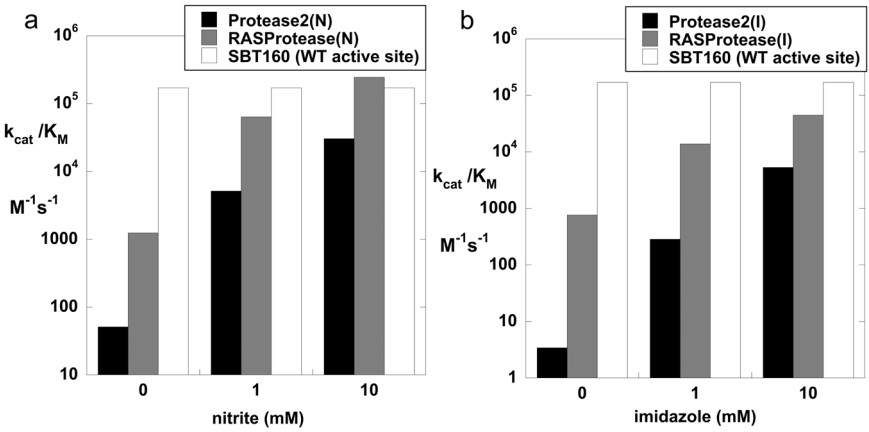

**Fig. 3 $k_{cat}/K_M$ for the target substrate QEEYSAM-AMC as a function of cofactor. a** Protease2(N) and RASProtease(N) vs. nitrite concentration. **b** Protease2(I) and RASProtease(I) vs. imidazole concentration. The activity of a progenitor protease (SBT160) with a complete catalytic triad (D32, H64, S221) is shown for comparison.

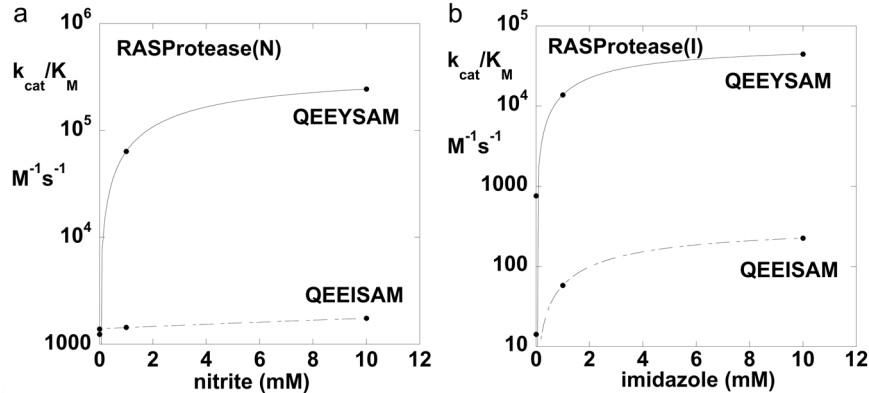

**Fig. 4 Conformational and chemical rescue by native and near-native sequences. a** $k_{cat}/K_M$ as a function of nitrite concentration for RASProtease(N) with the substrates QEEYSAM-AMC and QEEISAM-AMC. (**b**) $k_{cat}/K_M$ as a function of imidazole concentration for RASProtease(I) with the substrates QEEYSAM-AMC and QEEISAM-AMC.

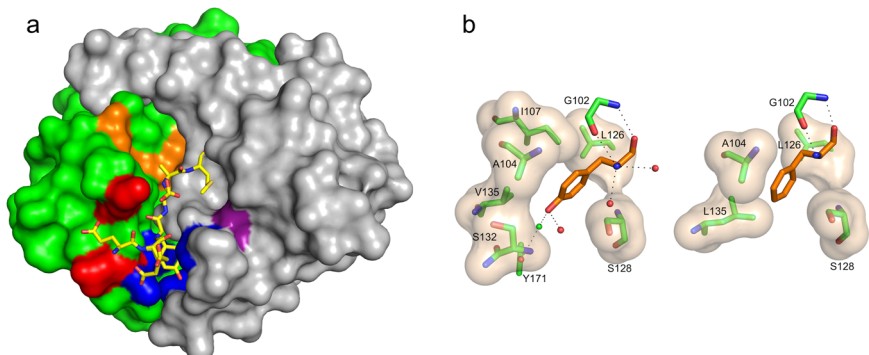

**Fig. 5 Engineering a protease directed against active RAS. a** Surface representation of RASProtease(I) with substrate binding sites colored purple (S1), orange (S2 and catalytic residues), red (S3), and blue (S4) with the bound YSAM peptide overlaid. **b** Interactions at the P4 site of RASProtease(I) (left) and its progenitor Protease1(N) (right). The van der Waals interactions between residues in the protease active site (carbons colored green) and the bound P4 residue (carbons colored orange) are represented as a tan surface. Hydrogen bonds are represented as black dashed lines.

and cofactor activation, we determined crystal structures of RASProtease(I) alone, in complex with YSAM and QEEYSAM product peptides, and with imidazole (Fig. 5a, S5–S8). We compared these structures with the crystal structure of Protease1(N) in complex with a peptide corresponding to its sequence specificity (LFRAL, Fig. 5b, right).

Overall the structures are very similar, with an RMSD between Cα carbons of 0.17 Å. The aromatic ring of the P4 side chain has common van der Waals interactions: the Cα and Cβ atoms of A104 interact with Cε1, the Cδ1 of L126 interacts with Cδ2, and Cβ of S128 interacts with Cδ2 and Cε2. Because of the space created in the S4 pocket by the L135V mutation, either a water or

an ion coordinates the hydroxyl of P4 Tyr, the hydroxyl of Y171, and the backbone nitrogen of S132 in RASProtease(I) (Fig. 5b, Fig. S5). We currently have this modeled as a chloride ion based on the peak height observed in an anomalous difference fourier map; however, added chloride does not appear to influence kinetic properties. P4 interactions are facilitated by a significant shift of the loop containing residues 130 to 133 of RASProtease(I) relative to the Protease1(N) complex. In fact, these are the only residues in the entire structure whose Cα positions shift by more than 1 Å, including the N- and C-termini.

In contrast to P4, the changes to the P3 site are more limited (Fig S6). Mutation of S101K increases activity with Ser at the P3 position. Hydrogen bonding interactions between the backbone nitrogen and carbonyl oxygen of the P3 residue and the corresponding partners on residue G127 are conserved.

The structures also provide insight into the structural basis for cofactor activation by both nitrite in Protease1(N) and imidazole in RASProtease(I). In both cases, a catalytic amino acid is mutated to Gly and the vacated space is occupied by a network of conserved water molecules in the absence of the cognate cofactor. Nitrite was modeled into the solvent network of the Protease1(N) structure (replacing HOH 54; Fig. S7A) such that it supplies the critical H-bond to the catalytic H64 and is coordinated to N33 and three conserved waters (23, 302, and 382). The coordination sphere of nitrite in Protease1(N) appears to be more complete than observed for azide in the parent enzyme (3BGO) and suggests why nitrite is tightly bound in spite of its lower pKa (3.37, compared to 4.72 for azide). Structures of RASProtease(I) without imidazole have three conserved waters (19, 81, and 122) that interact with Oδ1 and Oδ2 of D32, CO of S125, NH of G64, Oγ of S62, and Oγ of S221. When imidazole binds, these waters are displaced, the imidazole nitrogens H-bond to Oδ1 and Oδ2 of D32 and Oγ of S221, and the charge relay system is reconstituted (Fig. S8, S7B). An additional interesting feature of the structure of RASProtease(I) with QEEYSAM was the presence of an acyl adduct between the C-terminus of the P1 Met of the peptide and the Oγ of S221 of the enzyme (Fig S5). Apparently, the binding energy of the QEEYSAM peptide is sufficient to push its terminal carbon and Oγ of S221 into an orientation that drives the equilibrium from a product complex to the acyl enzyme[39]. Studying these types of subtle changes to the water structure in the active site that occur upon substrate and cofactor binding will allow us to make further changes to promote chemical and conformational rescue.

**Cleavage of RAS(GDP) and RAS(GMPPNP).** The next step in assessing our RAS-specific proteases was to monitor cleavage of native RAS protein, wherein switch 2 would adopt a range of conformations from extended to helical. Moreover, measuring protein cleavage allowed us to assess the relative rates of cleavage for active versus inactive RAS, thereby testing our hypothesis that active RAS is more vulnerable to proteolytic attack due to increased dynamic motion in switch 2. To trap RAS in an active conformation, we used an adduct with guanosine 5'-[β,γ-imido] triphosphate (GMPPNP), a slow hydrolyzing analog of GTP. The cleavage of RAS(GMPPNP) is then compared with the inactive RAS(GDP) form. The goal of the kinetic experiments is to achieve a quantitative understanding of individual steps in the reaction pathway. A minimum realistic mechanism is as follows:

$$ R + P \underset{\Longleftarrow}{\overset{K_S}{}} RP \overset{k_2}{\to} Pr_1 + r_2 \underset{\Longleftarrow}{\overset{K_P}{}} P + r_1 + r_2 \tag{1} $$

where R = RAS, P = protease, $r_1$ = the RAS N-terminal cleavage product and $r_2$ = the RAS C-terminal cleavage product. $K_S$ is the dissociation constant for uncleaved RAS and $k_2$ is the rate of the first chemical step in peptide bond cleavage (acylation). $K_P$ is the dissociation constant for the $r_1$ product fragment. To accurately determine kinetic parameters for RAS cleavage, we measured the kinetics of QEEYSAM-AMC cleavage in the presence of RAS(GDP) and RAS(GMPPNP). Data for RASProtease (I) are shown in Fig. 6a, b. The concentration of RAS(GMPPNP) or RAS(GDP) was varied from 0 to 20 μM. Interaction of protease with RAS is manifested as inhibition of peptide-AMC cleavage by RAS and RAS cleavage products. Early in the progress curves, the major inhibitory species is native RAS. As the reaction progresses, inhibition increases as the concentration of the $r_1$ fragment increases. $K_P$ of $r_1$ was determined in an independent series of experiments. To accomplish this, a fragment corresponding to the N-terminal cleavage product of RAS (amino acids 1–67, ending at YSAM) was expressed in E. coli and purified. Inhibition of protease by $r_1$ was measured by varying its concentration from 50 nM to 3.3 μM in reactions of each protease and with 0.1, 0.5, and 1 μM QEEYSAM-AMC (Table S2). NMR analysis of cleaved RAS shows that both $r_1$ and $r_2$ are disordered (Fig. S9). Kintek Explorer was used to fit all data to mechanism 1 (Fig. 6, Table S2). Values of $k_2/K_S$ were calculated to compare specificity for active RAS, independent of the $r_1$ product dissociation rate (Table S2). The analysis shows that $k_2/K_S$ for RASProtease(I) in 1 mM imidazole is $13,720 \, M^{-1} s^{-1}$ for RAS(GMPPNP) and $220 \, M^{-1} s^{-1}$ for RAS(GDP) (60-fold preference for the active form). Similar measurements and global fits also were made for RASProtease(N) in 1 mM nitrite (Fig. 6c, d): $k_2/K_S$ is $32,430 \, M^{-1} s^{-1}$ for RAS (GMPPNP) and $410 \, M^{-1} s^{-1}$ for RAS(GDP) (80-fold preference for the active form).

Although less quantitative, the increased rate of cleavage of RAS(GMPPNP) relative to RAS(GDP) could also easily be observed by SDS gel analysis (Fig. S10). Analysis by MALDI confirms that the enzyme cleaves RAS after the QEEYSAM site with no detectible off-target cleavages (Fig. S11). Consistent with the low intrinsic activity observed in peptide assays, no cleavage of RAS was observed in the absence of a cognate cofactor with either enzyme. Measurements of binding uncleaved RAS (GMPPNP) and RAS(GDP) with RAS protease were also made by gel filtration and show the increased affinity of protease for RAS(GMPPNP) (Fig. S12).

**Analysis of the target region in RAS by NMR.** To better understand how dynamics in switch 2 contribute to RAS protease specificity for the active form, we examined the dynamics of this region by NMR. An order-to-disorder transition of switch 2 has been previously observed in RAS crystal structures[40,41]. Two-dimensional $^1H$-$^{15}N$ HSQC spectra indicate extensive structural differences between the GDP- and GMPPNP-bound forms of RAS (Figs. S13A, B). NMR backbone resonance assignments were made using standard procedures and deposited in BioMagResBank (accession codes for the GDP and GMPPNP forms are 28008 and 28009, respectively). Most of the amide signals in the switch 1 (residues 30–38) and switch 2 (residues 59–76) regions are detectable in the GDP-bound state but are exchange broadened in the GMPPNP-bound form, indicating enhanced dynamics in the GMPPNP state on the NMR timescale (microseconds to milliseconds). Comparable observations have been made with WT HRAS, WT KRAS, and other RAS mutants[42–49]. Analysis of differences in backbone dynamics using $^{15}N$-relaxation measurements indicates that the GMPPNP-bound form of RAS is also considerably more flexible on the ps-ns timescale than the GDP-bound form (Fig. S13C, Fig. S14). Thus, the effect of GMPPNP binding is to increase main chain flexibility for a large number of residues in the molecule over a wide timescale range, making the GMPPNP- and GTP-bound states, and the switch regions in particular, more susceptible to proteolytic cleavage than the GDP-bound form.

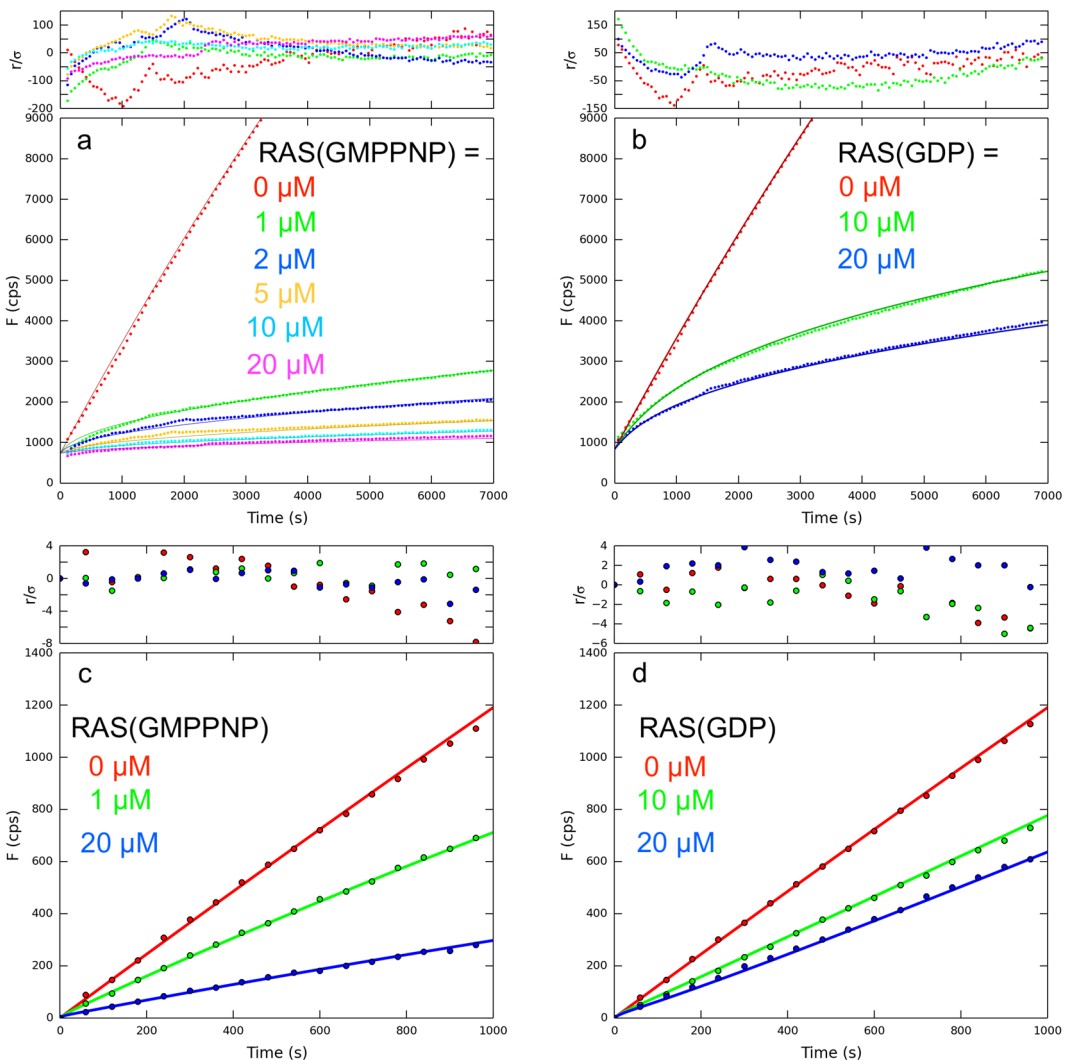

**Fig. 6 Kinetics of AMC release from QEEYSAM-AMC by 100 nM RASProtease in the presence of RAS. a** RASProtease(I) + RAS(GMPPNP).
**b** RASProtease(I) + RAS(GDP). (**a**) and (**b**) were measured in the presence of 1 μM QEEYSAM-AMC and 1 mM imidazole. **c** RASProtease(N) + RAS
(GMPPNP). **d** RASProtease(N) + RAS(GDP). (**c**) and (**d**) were measured in the presence of 1 μM QEEYSAM-AMC and 1 mM nitrite. Data points are solid
circles. Global fit to mechanism 1 are solid lines. Residuals are plotted above each graph.

**Co-expression of RAS with RASProtease(I) in *E. coli*.** To
measure activity and specificity in cells, we developed a bacterial
system for co-expressing RAS with RAS-specific proteases. We
constructed genes for RAS fusion proteins (Fig. S15) that con-
sisted of an N-terminal $G_A$ domain[50,51], amino acids 1-166 of
human HRAS, and a C-terminal cellulose binding domain[52]. The
small N-terminal and C-terminal binding domains allow easy
purification of the entire fusion protein as well as both cleavage
products so that the precise cleavage site in *E. coli* can be deter-
mined. The expression of the RAS fusion protein is shown in Fig.
S16. The protease was co-expressed as a zymogen (34.9 kDa)
consisting of an N-terminal inhibitory (I) domain (8.5 kDa) and
the mature protease domain (26.4 kDa)[53]. Growth was at 37 °C
either without imidazole (Fig. S17) or with 100 μM imidazole
(Fig. 7) added to the culture media. The gel patterns show that the
protease zymogen is processed into the I-domain and mature
protease in both cases, but RAS cleavage is dependent on imi-
dazole and coupled to degradation of the I-domain into smaller
fragments. RAS fragments $r_1$ and $r_2$ appear as intact I-domain

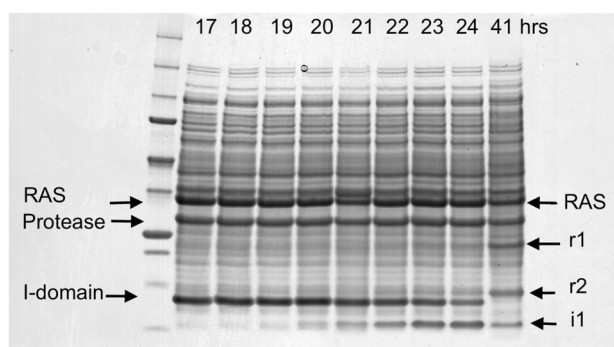

**Fig. 7 RASProtease(I) with 100 μM imidazole added to the culture media
at 17 h.** Intact RAS fusion protein is 35,974 daltons. The N- and C- terminal
fragments of RAS are 13,579 (r2) and 22,413 daltons (r1), respectively. The
C-terminal fragment of the I-domain is 6,175 daltons (i1). Markers: 250,
150, 100, 75, 50, 37, 25, 20, 15, 10 kDa.

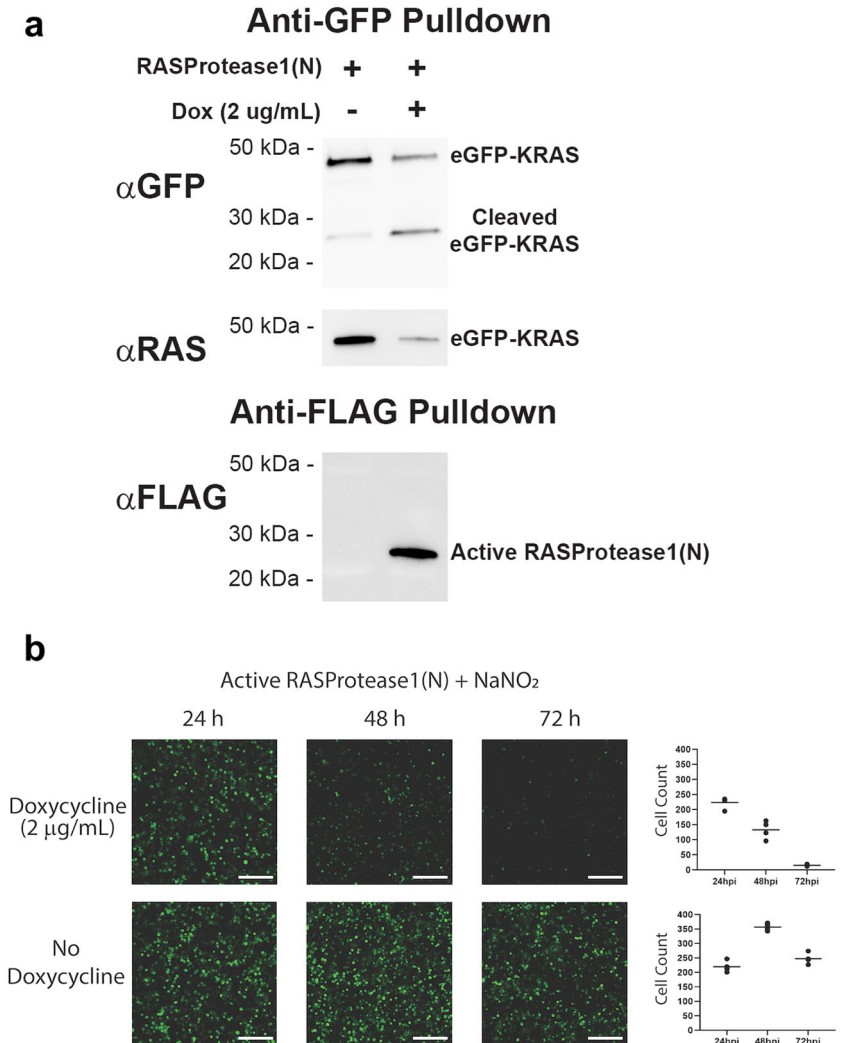

**Fig. 8 RAS-specific protease activity in cells. a** Western blot analysis of cells co-transfected with eGFP-KRAS and RASProtease shows the appearance of a KRAS cleavage product upon induction of the active protease when probed with an anti-GFP antibody following a GFP pull-down. Sodium nitrite was added to the cell culture medium at a final concentration of 1 mM to mitigate potential variability in cellular nitrite concentrations. Appearance of this product coincides with depletion of a RAS-reactive band when probed with an anti-RAS antibody. Appearance of cleaved eGFP-KRAS also coincides with expression of activated protease that has cleaved its inhibitory I-domain. **b** Induction of the active protease in HEK 293 T cells at 24 h after transfection with nitrite supplemented culture medium results in a marked decrease in GFP fluorescence at 48 and 72 h after transfection compared to the same cells without induction of protease expression. The scale bar is 200 μm.

disappears. By 41 hrs. in imidazole, all I-domain is cleaved and greater than 50% of RAS is cleaved (Fig. 7). The gel pattern also shows that RAS is specifically cleaved into two discrete fragments. We confirmed that RAS was cleaved after the QEEYSAM sequence by purifying the $r_1$ fragment from the *E. coli* extract and measuring its mass by MALDI. There is no indication in the gel pattern of *E. coli* proteins being degraded. The fact that ~50% of RAS remains intact after 41 h is consistent with a high RAS protease preference for active RAS in the cell. Newly synthesized RAS predominantly binds GTP and therefore initially exists in the dynamic, active conformation. RAS(GTP) then converts to RAS(GDP) at a rate of ~1 h$^{-1}$ [54–56]. Because *E. coli* lacks GDP-GTP exchange factors, we assume that RAS remains in the GDP bound form after hydrolysis and is less vulnerable to cleavage. After new RAS synthesis stops, RAS accumulates in the inactive and partially protease-resistant form.

**Co-expression of RAS with RASProtease(N).** Fig. S18 shows co-expression of the RAS gene with the RASProtease(N) zymogen

gene. RASProtease(N) cleaves ~70% of RAS into two discrete fragments within 18 h of growth but the remaining RAS remains intact after 48 h. Unlike the case with imidazole, we are not able to precisely control nitrite concentration because it is an intermediate in metabolic pathways involving nitrogen in *E. coli* [57,58]. Nevertheless, the RAS cleavage by nitrite-activated proteases is potentially informative because nitrite is a disease marker in eukaryotic cells and its concentration in *E. coli* is likely similar to that in cancer cells [57,58]. RASProtease(N) appears to be more active in *E. coli* with endogenous nitrite than RASProtease(I) in 100 μM imidazole. Even so, a cleavage-resistant population of RAS remains after 48 h. We presume this is RAS(GDP).

**Engineered proteases can destroy active RAS in mammalian cells.** Finally, we tested whether a RASProtease(N) could cleave the switch 2 target sequence in a human cell. Based on our NMR data and the fact that switch 2 is conserved among the 3 major RAS isoforms, any RAS isoform would be appropriate for this type of experiment. However, since KRAS is the most important

cellular and therapeutic target, we used KRAS as our target for cell-based experiments. We employed fluorescently tagged RAS (eGFP-KRAS) as a means to simplify the detection of cleaved product. This fusion (48 kDa) can be observed in a Western blot probed with an anti-GFP monoclonal antibody (Abcam, Fig. 8a, lane 1). Induction of expression of the protease zymogen with doxycycline results in a marked depletion of eGFP-RAS and a corresponding increase in the presence of a band consistent with the eGFP protein from the fusion (Fig. 8a, lane 2). Probing the same samples with an anti-RAS antibody (Ras10, ThermoFisher) confirms that this band contains an eGFP-RAS fusion and further shows that RAS disappearance coincides with the appearance of the eGFP product (Fig. 8a, lanes 1 vs. 2). As expected, primary processing of the RASProtease(N) zymogen (34.9 kDa) to the mature protease (26.4 kDa) occurs readily in cells (Fig. 8A, lane 2). In addition, fluorescence microscopy images show a marked decrease in eGFP signal upon induction of RASProtease(N) (Fig. 8b). Disappearance of the eGFP-RAS fragment indicated that, as in *E. coli*, cleaved RAS is further degraded by cellular proteases. As expected, western blot analysis of cells co-transfected with eGFP-KRAS and inactive (S221A) RASProtease does not result in the appearance of a KRAS cleavage product upon induction of the inactive protease when probed with an anti-GFP antibody following a GFP pull-down (Fig. S19A). Likewise, the intensity of the RAS-reactive band remains constant when probed with an anti-RAS antibody. The inactive protease fails to cleave its inhibitory I domain as evidenced by the unprocessed protease band at approximately 35 kDa (Fig. S19A). Induction of the inactive protease in HEK 293T cells at 24 h after transfection results in no change in GFP fluorescence at 48 and 72 h after transfection compared to the same cells without induction of protease expression (Fig. S19B).

## Discussion

Our goal in this work was to develop principles for engineering protein-specific proteases and to apply these principles to target active RAS. The key to engineering high sequence specificity in a protease is linking binding at sub-sites with chemical steps in peptide bond hydrolysis. This was accomplished by exploiting two facts about enzymes: (1) mutations at remote sub-sites can affect the conformation of catalytic amino acids; (2) change in the conformation of the catalytic region affects chemical rescue of active site mutants. High-specificity occurs when cognate sequence binding at sub-sites is compatible with productive binding of cofactor but binding of incorrect sequences antagonizes cofactor binding.

To understand substrate and chemical rescue, it is useful to consider some basic structural features of subtilisin proteases. Subtilisin has a cardioid shape with the active site and substrate

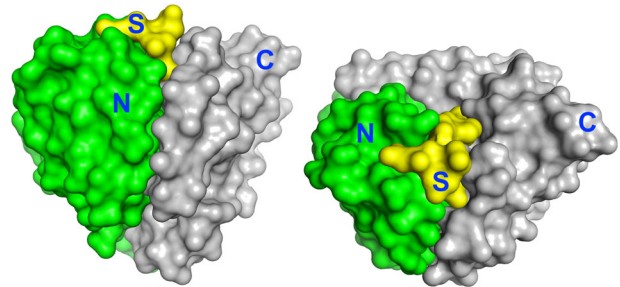

**Fig. 9 Two views of the domain structure for RAS-specific subtilisin (6UAO.pdb) with substrate bound.** The N-terminal domain (N) is green, C-terminal domain (C) is gray, and QEEYSAM substrate (S) is yellow.

binding pockets forming a cusp that divides N-terminal and C-terminal domains (Fig. 9, [28]). The catalytic D32 and H64 and the P2, P3, and P5 sub-sites are primarily associated with the N-terminal domain (Fig. 5a). The catalytic S221 and P1, P4, and P6 sub-sites are primarily associated with the C-terminal domain. Thus the substrate intercalates between the two domains, bridges the interface, and affects the association between the two. Subtilisin specificity, in general, can be understood in terms of a model in which the domain interface is either in a deformed, low-activity conformation or in a canonical active conformation. When the domain interface is stable, the substrate adapts to the enzyme and specificity is broad. This has also clearly been shown for α-lytic protease which resists deformation of catalytic amino acids even as binding pockets conform to bind a range of substrate sequences[59]. When the stability of the domain interface of a protease is decreased by mutation, however, the enzyme conforms to the substrate and activity will be low unless the substrate fit is precise and promotes the active conformation (i.e. in part by reestablishing the native interface between domains). Because individual sub-sites and the catalytic triad are interconnected, distortion in one area affects the other. Rheinnecker et al.[9,11] have shown previously that certain mutations in the S4 pocket of subtilisin, particularly those that form cavities, adversely affect activity of even small substrates that do not interact at S4. This led to an increase in specificity for peptide substrates with L or F over A at P4. To create RAS-specific proteases, we combined traditional sub-site engineering approaches with mutations that destabilize the domain interface. The best mutations weaken the interface between the two domains but generate favorable interactions with a cognate substrate and/or the cofactor. The favorable interactions rescue the active site conformation at the interface resulting in high activity. For example, removing a catalytic residue (e.g. D32G or H64G) reduces activity both by eliminating an element of the charge relay system and weakening the domain interface, but also creates potential for rescuing the active conformation and chemistry by the cognate cofactor and substrate. Likewise, enlarging the S4 pocket to accommodate tyrosine causes instability that is transmitted to the weakened catalytic site. Binding of the cognate substrate with a P4 tyrosine rescues the active conformation, however, by supplying stabilizing interactions in the S4 pocket that are transmitted to the catalytic region and promote productive binding of cofactor. The magnitude of cofactor activation depends on the population of the active conformation in the apo enzyme relative to its population with substrate and cofactor bound. This is a delicate balance. If the domain interface is too stable, a mutant will be fast but non-specific. If the interface is too unstable, however, a mutant will be specific but slow. To produce high specificity and activity, the energies of the inactive and active conformations must be close enough that cognate binding to sub-sites significantly populates the active form[60]. This creates a critical state in which both cofactor binding and cognate sub-site interactions become required for activity. These proteases have very low activity against all substrates in the absence of cofactor but allow rescue of the active conformation by the cognate sequence with its cofactor.

Comparing several high-resolution structures reveals structural changes that may contribute to the ability of cognate substrate and cofactor binding to rescue activity in RAS-specific proteases. RAS-specific proteases have an expanded S4 site that binds numerous solvent molecules in the enlarged S4 pocket. Structures with a cognate peptide bound show that the L135V mutation allows space for an adventitious ion-binding site that coordinates the hydroxyl group of the P4 tyrosine of the substrate. This stabilizing interaction may be an important element for rescuing the active conformation of the catalytic region and enabling strong cofactor activation. Likewise, mutating a catalytic amino acid to

Gly creates space that is occupied by a network of conserved waters in the absence of the cognate cofactor.

It might be assumed that enzymes engineered in this way would be slow[61]. This is not true. Kinetic analysis shows proteases targeting RAS are both more specific and more active than the viral processing proteases TEV and HRV 3C are for their cognate sequences[62,63]. For example, in the presence of 1 mM cofactor, RASProtease(N) and RASProtease(I) have $k_{cat}/K_M$ of > $10^4$ M$^{-1}$ s$^{-1}$ compared to $k_{cat}/K_M$ values of ~$10^3$ M$^{-1}$ s$^{-1}$ for 3C-type proteases.

Engineering cofactor activation also proved to be useful for regulating protease activity inside cells. Imidazole is benign but normally not present inside cells. Thus, it can be used as a xenobiotic activator to tightly control RAS cleavage in *E. coli*. Controlling RAS cleavage in *E. coli* with nitrite is more complicated because it is an intermediate in metabolic pathways involving nitrogen. Nitrite may be quite useful for regulating activity in eukaryotic cells, however. Elevated nitrite is a common signature of disease states (including RAS-related cancers)[64–66] and reaches concentrations >100 μM in tumor cells[67,68]. This occurs because disease-induced nitric oxide synthase produces nitric oxide (NO) and NO quickly oxidizes and accumulates as nitrite. To date, we have only tested one RAS protease in human cells, but we have established that a RAS-specific protease can self-activate in human cells, locate KRAS at the plasma membrane, and cleave it as indicated by the presence of the eGFP fusion product and the precipitous disappearance of KRAS (Fig. 8). Although RAS isoforms are highly abundant in human cells (~300,000 RAS molecules (150 nM) in colorectal cancer cells)[69], the proteases developed in *E. coli* will need to be adapted to the new environment if precise control of RAS signaling is the goal. Perhaps the more important point, however, is that the general principles learned from *E. coli* makes re-programming cultured human cells appear feasible and may provide strategies to selectively cleave RAS in cancer cells.

Primary structure alone is generally insufficient for encoding protease specificity. For example, Caspases are considered highly specific, but it is not possible to predict their natural protein targets from their cleavage patterns on small peptides[70,71]. Thus, a critical element of specificity is discrimination between different conformations of the same sequence. To use the conformation of the target protein to increase specificity, we chose a target sequence that is partially exposed in active RAS but is typically found in amphipathic α-helices and resistant to proteolysis when it occurs in other proteins. NMR analysis indicated extensive structural changes between the GDP- and GMPPNP-bound forms of RAS and increased mobility of the QEEYSAM sequence in RAS(GMPPNP). Specificity is governed therefore by both the correct primary structure and dynamic changes in the secondary structure of the target region. The additional information from conformation allows much higher specificity than can be achieved based on sequence alone. The effectiveness of this type of recognition is manifested in the 60 to 80-fold difference in cleavage rate between active and inactive RAS. In this sense inactive RAS serves as an internal control. To be useful in a cell, a protein-specific protease must selectively destroy the target protein and not many competing substrates. Experiments with RAS-specific proteases in cells show this to be the case. Significant depletion of RAS can be achieved without any apparent effect on cell viability or noticeable degradation of endogenous proteins.

In conclusion, the principles presented here are general and can be applied to many target proteins. This includes proteins involved in aberrant signal transduction but also includes foreign proteins involved in cell invasion. The process of creating new protein-specific proteases begins with matching the specificity of an existing protease with changes in local or global stability in a desired target protein. It culminates with designing-evolving the protease to match the new target sequence and cofactor environment.

## Materials and methods

**Mutant subtilisins**. SBT189 (3BGO.pdb) is our starting subtilisin and denotes subtilisin from *Bacillus amyloliquefaciens* with the following mutations: Q2K, S3C, P5S, S9A, I31L, D32A, K43N, M50F, A73L, Δ75-83, Y104A, G128S, E156S, G166S, G169A, S188P, Q206C, N212G, Y217L, N218S, T254A, Q271E.[21,22] Expression carried out in *E. coli* by auto-induction[72] and purification was by affinity chromatography using a cognate 7-mer peptides purchased from AnaSpec, Inc.[73,74].

**Kinetic measurements**. Synthetic peptide-AMC (7-amino-4-methylcoumarin (AMC, Ex: 350 nm, Em: 450 nm) was purchased from AnaSpec Inc. Concentrations of the AMC substrates were determined by absorbency at 324 nm using an extinction coefficient of 16 mM$^{-1}$cm$^{-1}$. Reaction kinetics of AMC substrates were measured using a KinTek Stopped-Flow Model SF2001 (Ex: 380 nm, Em: 400 nm cutoff filter). Kinetic data were fit using KinTek Global Explorer software obtained from the KinTek Corporation website (www.kintek-corp.com). Kinetic measurements of longer reactions, after manual mixing, were determined using a BioTek Synergy HT plate reader.

**Statistics and reproducibility**. Highly pure (≥98%) protease was used for all kinetic experiments. Kinetic parameters were determined using at least five substrate concentrations for each experiment. Determinations of were made at least three times for each substrate-enzyme-cofactor combination. Deviations in $k_{cat}/K_M$ were < 5% in independent determinations.

**Protein expression and purification of RAS**. The genes for human HRAS (amino acids 1-166) were cloned into the vector pPal8[16], which encodes an engineered subtilisin pro-sequence as an N-terminal fusion domain. The resulting fusion proteins were produced in *E. coli* and purified using an affinity-cleavage tag system, which we developed[16,75]. A commercial version of the purification system is available through Bio-Rad Laboratories (Profinity eXact Purification System). Exchange of GDP in recombinant preps[76] for GMPPNP was performed using EDTA to accelerate nucleotide dissociation and exchange[77].

**Cell lines and plasmid constructs**. HEK 293T cells[78] were purchased from ATCC. ORFs for protease clones were synthesized by Genscript (Piscataway, NJ) and subcloned into pLVX-TetOne-Puro. The eGFP-KRAS plasmid was provided by the RAS Initiative (Frederick, MD).

**Analysis of RAS cleavage in cells**. HEK 293T cells were seeded into 6-well plates at a density of $1.2 \times 10^6$ cells per well in DMEM plus 10% Tetracycline-free FBS. Sixteen hours after seeding, cells were transfected with 1.25 μg of each plasmid using Lipofectamine 3000 (Thermo Fisher) according to the manufacturer's protocol. Transfected cells were incubated for 24 hr at 37 °C in a humidified incubator providing 5% $CO_2$. Twenty-four hours after transfection, expression of the protease was induced by addition of doxycycline to fresh media at a final concentration of 2 μg/mL, followed by incubation at 37 °C for an additional 48 hr.

**NMR spectroscopy**. NMR spectra of RAS-GDP and RAS-GMPPNP were recorded on a Bruker Avance III 600 MHz spectrometer fitted with a cryoprobe. Samples for three-dimensional experiments were approximately 0.25 mM RAS in 20 mM HEPES, 50 mM NaCl, 5 mM MgCl₂, and 1 mM TCEP, pH 7.4. All assignment experiments were collected at 25 °C. The following standard three-dimensional heteronuclear experiments were acquired: HNCACB, CBCA(CO)NH, HN(CA)CO, HNCO. Triple resonance experiments were performed with 25% non-uniform sampling[79]. Amide chemical shift perturbations between the GDP and GMPPNP-bound forms of G12V-HRAS, $\Delta\delta_{total}$, were calculated from the equation $\Delta\delta_{total} = [(W_H\Delta\delta_H)_2 + (W_N\Delta\delta_N)_2]^{1/2}$, where $\Delta\delta_H$ and $\Delta\delta_N$ are the proton and nitrogen shift differences for a given resonance, respectively, and $W_H = 1$ and $W_N = 0.2$ are weighting factors. Two-dimensional heteronuclear {¹H}-¹⁵N steady-state NOE experiments were collected with a gradient-selected, sensitivity-enhanced pulse sequence[80]. Spectra were recorded with and without saturation in an interleaved manner employing a 5 s recycle delay. The heteronuclear NOEs were calculated from the NOE_on/NOE_off ratio and standard deviations were obtained from measured background noise levels. ¹⁵N $R_1$ and $R_2$ measurements were carried out using gradient-selected, sensitivity-enhanced 2D ¹H-¹⁵N HSQC experiments[80] with water flip-back modifications for solvent suppression. ¹⁵N $R_1$ experiments were acquired with variable delay times of 10, 150, 300, 400, 600, 900, and 1200 ms. ¹⁵N $R_2$ experiments were acquired with delay times of 8, 16, 24, 32, 40, 48, 64, 80, 96, and 130 ms. $R_1$ and $R_2$ values were obtained from single exponential decay fitting with error estimates for $R$ using Sparky. Generalized order parameters ($S^2$) were extracted from the ¹⁵N relaxation data utilizing the Modelfree program[81]. Spectra were processed using NMRPipe[82] and analyzed with Sparky[83].

**Table 1 Data collection and refinement statistics (molecular replacement).**

| | RASProtease(I) | RASProtease(I) + YSAM | Protease1(N) + LFRAL | RASProtease(I) + QEEYSAM |
|---|---|---|---|---|
| *Data collection* | | | | |
| Space group | P4₁2₁2 | P4₁2₁2 | P4₁2₁2 | P4₁2₁2 |
| *Cell dimensions* | | | | |
| *a, b, c* (Å) | 58.65, 58.65,124.75 | 58.64, 58.64, 125.17 | 58.65, 58.65, 125.67 | 58.51, 58.51, 125.09 |
| α, β, γ (°) | 90, 90, 90 | 90, 90, 90 | 90, 90, 90 | 90, 90, 90 |
| Resolution (Å) | 53.3–1.70 (1.73–1.70)ᵃ | 53.1–1.20 (1.22–1.20) | 19.3–1.60 (1.66–1.60) | 42.7–1.63 (1.66–1.63) |
| $R_{meas}$ | 0.099 (0.424) | 0.105 (0.120) | 0.041 (0.158) | 0.093 (0.808) |
| $I/\sigma(I)$ | 26.7 (8.3) | 30.6 (25.4) | 25.2 (7.6) | 20.6 (2.8) |
| Completeness | 100.0 (100.0) | 100.0 (100.0) | 98.9 (91.3) | 100 (99.8) |
| Redundancy | 27.2 (26.6) | 22.6 (22.6) | 5.8 (3.2) | 13.5 (13.2) |
| *Refinement* | | | | |
| Resolution (Å) | 53.08–1.70 (1.74–1.70) | 41.5–1.20 (1.23–1.20) | 19.3–1.60 (1.64–1.60) | 41.4–1.63 (1.67–1.63) |
| $R_{work}/R_{free}$ (%) | 14.7/17.4 | 10.9/12.5 | 12.8/15.8 | 14.9/18.0 |
| *No. atoms* | | | | |
| Protein | 1854 | 1976 | 1943 | 1940 |
| Ligand/ion | 30 | 31 | 32 | 22 |
| Waters | 185 | 321 | 299 | 242 |
| *B factors* (Å²) | | | | |
| Protein | 13.2 | 7.2 | 13.0 | 13.9 |
| Ligand/ion | 33.9 | 24.9 | 31.6 | 29.7 |
| Waters | 23.5 | 23.3 | 28.7 | 26.1 |
| *R.m.s. deviations* | | | | |
| Bonds (Å) | 0.02 | 0.02 | 0.02 | 0.01 |
| Angles (°) | 2.1 | 2.2 | 2.0 | 1.8 |

Each structure was determined from a single crystal
ᵃValues in parentheses are for the highest resolution shell

**Crystallization, data collection, and processing**. Purified RASProtease(I) was concentrated to 7 mg/mL for use in crystallization screening. The best crystals from our screens were obtained in a condition containing 0.1 M Bis–TRIS propane pH 8.5, 0.2 M KSCN, and 20% PEG 3350. Crystals appeared overnight and grew to a maximum size after 2–3 days. These crystals belong to space group P4₁2₁2 and have unit cell dimensions $a = b = 58.6$ Å, $c = 124.8$ Å, α=β=γ=90°. Native data up to 1.7 Å resolution were collected at 100 K using in-house X-ray diffraction resources at a wavelength of 1.5418 Å.

For the RASProtease(I)-YSAM complex, RASProtease(I) crystals were soaked overnight in mother liquor supplemented with the YSAM peptide at a final concentration of 2.1 mM. RASProtease(I) crystallizes in a variety of conditions and for the YSAM complex, the mother liquor for the YSAM complex contained 0.1 M TRIS-HCl pH 8.0, 20% PEG 6 K, 0.2 M NaCl. Crystals were transferred to the crystallization condition supplemented with 15% glycerol and 2.1 mM YSAM peptide and flash frozen in liquid nitrogen prior to data collection. Native data up to 1.2 Å resolution were collected at Stanford Synchrotron Radiation Laboratory (SSRL) beamline 12-2. The data were reduced using mosflm[84] and Aimless[85] from the CCP4 program suite[85]. A similar procedure was used to generate the RASProtease(I) -QEEYSAM complex. The mother liquor for these crystals was 0.1 M HEPES pH 7.0, 20% PEG 6 K, 0.2 M NaCl. Data for the RASProtease(I)-QEEYSAM complex were collected to 1.63 Å using in-house X-ray diffraction resources.

Similarly, purified Protease1(N) was concentrated to 12 mg/mL in a buffer composed of 5 mM HEPES pH 7.0, 10 mM Azide, and 0.53 mM LFRAL pentapeptide. Initial crystallization screening at 17 °C using in-house resources identified a number of potential conditions, the best of which was 30% PEG 8 K, 0.2 M NaCl, and 0.1 M Imidazole pH 8.0. This condition was subsequently refined to 0.1 M Imidazole pH 8.4, 0.2 M NaCl, and 25% PEG 8 K. Like the RASProtease(I) crystals, these crystals belong to space group P4₁2₁2 and have unit cell dimensions $a = b = 58.7$ Å, $c = 125.7$ Å, α=β=γ=90°. A crystal from this condition was cryoprotected using mother liquor mixed with glycerol to give a final glycerol concentration of 17%, cryocooled directly in the gaseous nitrogen stream of the X-ray source, and diffraction data were collected up to 1.6 Å resolution. The data were reduced using the D*Trek package[86].

**Structure determination, model building and refinement**. Initial phases for RASProtease(I) and the RASProtease(I) -YSAM complex were determined by molecular replacement using the program Molrep[87] using the coordinates of the subtilisin chain in a subtilisin-prosegment complex structure (PDB ID: 1SPB). Prior to molecular replacement, all solvent molecules and ions were removed from the search model. The structure was refined using Refmac5[88–92] from CCP4 program suite. Iterative cycles of model building using COOT[93–96] yielded structures with $R_{work}/R_{free}$ of 0.14/0.18 for RASProtease(I), 0.09/0.11, for the RASProtease(I)–YSAM complex, and 0.15/0.18 for the RASProtease(I)–QEEYSAM

complex. A summary of the refinement statistics for all structures is provided in Table S3.

Initial phases for the Protease1(N)–LFRAL complex were determined by using the program Molrep[87] using the coordinates of a previously determined subtilisin crystal structure (PDB ID: 3F49). Prior to molecular replacement, all solvent molecules and ions with B factors greater than 20 Å² were removed from the search model. The structure was refined using Refmac5[88–92] from the CCP4 program suite. Iterative cycles of model building using COOT[93–96] yielded a structure with $R_{work}/R_{free}$ of 0.16/0.18. A summary of the refinement statistics for the structure is provided in Table 1. The refined structure was deposited in the PDB (Accession Code: 6UBE).

**Analysis of KRAS cleavage in cells**. To ensure a relatively uniform amount of nitrite in cells transfected with the designed protease, 1 mM sodium nitrite was added to the media along with doxycycline. Following this incubation, the media in each well was replaced with PBS and the cells from each well were harvested. eGFP-KRAS and/or its cleavage products were isolated from the cells using a GFP-trap_A immunoprecipitation kit (Chromotek). Briefly, cells were lysed in 10 mM TRIS-HCl pH 7.5, 150 mM NaCl, 0.5 mM EDTA, 1% (v/v) Triton X-100 on ice according to the manufacturer's recommendations. After centrifugation to remove insoluble cell debris, the lysates were incubated with anti-GFP beads that had been equilibrated in 10 mM TRIS-HCl pH 7.5, 150 mM NaCl, 0.5 mM EDTA overnight with gentle agitation at 4 °C. The beads were then washed three times with 10 mM TRIS-HCl pH 7.5, 150 mM NaCl, 0.5 mM EDTA. Following removal of the wash buffer, the beads were resuspended in SDS-PAGE running buffer and boiled for 10 min at 95 °C prior to analysis by Western blotting. The same lysates were then used for anti-FLAG pulldowns using a FLAG immunoprecipitation kit (Sigma) according to the manufacturer's instructions. The wash buffer was 50 mM TRIS-HCl pH 7.5, 150 mM NaCl and the samples were boiled for three minutes at 95 °C prior to analysis by Western blotting.

**Microscopy**. Six-well tissue culture plates (Thermo Fisher Scientific cat. No. 140675) were imaged with a Zeiss LSM 710 AxioObserver microscope with plate adapter (cat. no. 451353-0000-000). Eight bit images were taken with an EC Plan-NeoFluar 10x/0.30 M27 objective. A 16% power, 514 nm wavelength was used to excite GFP conjugated to protein. Four images from each well near the center were taken to minimize reflection and light artifacts at 24, 48 and 72 h post-transfection resulting in 24 images at each timepoint.

**Quantifying fluorescent cells**. A representative timepoint image was opened in Zen Lite (Blue Edition v. 2.6, Zeiss). The green channel histogram was set from default 255 to 50. This increased the brightness before being exported into.tiff format. Images in the timepoint folder were processed similarly using "Batch"

mode. Representative images were opened in Photoshop (CC v. 20.0.0, Adobe). Each was converted to black and white. Green channel was set to 300%. Next the images were flattened and inverted. Exported images were grayscale, 8-bit,.tiff files after using "Actions" (batch processing) in Photoshop. Images were imported into FIJI[1] (Image J, NIH) and counted with the "Analyze Particles" module. Size was set to 0.01–0.04 with circularity equal to 0.00–0.50. The settings were saved to a "Macro" and used as a batch process on all images in the folder. Prism (v8.0, GraphPad) was used to plot the counts. Additional information can be accessed at: https://www.protocols.io/view/quantifying-fluorescent-cells-in-mammalian-cell-ti-y2nfyde.

**Reporting summary**. Further information on research design is available in the Nature Research Reporting Summary linked to this article.

## Data availability

The refined X-ray structures are available in the PDB (Accession Codes: 6U9L, 6UAI, 6UAO). The source data for the charts and graphs in the figures are available as Supplementary data 1. All other data are available from the corresponding authors on reasonable request.

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

## Acknowledgements

The authors thank Vanessa Wall, Dominic Esposito, and Frank McCormick (NCI-Frederick) for scientific guidance and reagents, including the eGFP-KRAS reporter plasmid. Dominic Esposito also critically read the manuscript and gave helpful suggestions. This work was supported by MPower-IBBR (University of Maryland) Seed Funds, R44CA163403, R44GM103389, R44GM126676, and R01GM062154. The NMR facility is jointly supported by the University of Maryland, the National Institute of Standards and Technology, and a grant from the W. M. Keck Foundation. Mention of commercial products does not imply recommendation or endorsement by NIST.

## Author contributions

Yw.C., E.T., and B.R. contributed equally. Protease engineering: Yw.C., B.R., E.C., D.R., R.S., P.B.; Performed protease characterization and kinetic analysis: B.R., Yw.C., R.S., P.B.; Performed *E. coli* cell assays of RAS cleavage: Yw.C., B.R; Performed NMR experiments/structural analysis: Yh.C., Y.H., J.O.; Performed X-ray crystallography/structural analysis: G.C., D.G., E.T.; Purified and characterized RAS-nucleotide complexes: R.R.; Performed mammalian cell assays of RAS cleavage: R.W., H.K., E.T.; Assisted in design of mammalian cell assays: M.S., S.M, R.R., D.W., T.F.; Wrote paper: J.O. (NMR analysis), E.T. (analysis of X-ray structures and mammalian cell expression), Yw.C., B.R., P.B. (remaining sections).

## Competing interests

The authors declare no competing interests.
