## [Peer Review File · Communications Biology]

Reviewers' comments:

Reviewer #1 (Remarks to the Author):

The manuscript by Chen et al describes a stunning feat of protein engineering: the creation of a protease that selectively cleaves the switch 2 region of HRAS. The experiments appear to be carefully performed, though poorly described in places. The paper also suffers from a shifting focus and poor organization. For example, the results section starts with an NMR analysis of G12V HRAS, suggesting that goal is to selectively degrade mutant RAS. However, G12V does not appear again until Figure 8. Lastly, the authors do not appear to have a coherent set of experiments- for example, the imidazole-regulated protease is tested in bacterial cells but not human cells while the nitrite-regulated protease is tested in human cells but not bacteria. It might be that the authors could compose a compelling paper that focused on one enzyme and the outcome (regulated RAS-selective cleavage) but did not include the problematic bits (e.g., quantitative analysis of RAS cleavage). Specific comments below.

Major Comments

1. The S4 site appears to have a crucial Cl ion that provides specificity for the P4 Tyr residue. Why do the authors believe this density is Cl ion? There does not appear to be any Cl in the crystallization buffer for SBT2208 (p22)- perhaps it is an unlisted counterion for the Bis-Tris propane? I could not find the buffer conditions for the protease assays- do they include Cl? Does selectivity depend on the presence of Cl? This is a crucial information for understanding the origin of specificity at the P4 site.
2. Figure 5. The authors have set up the idea that G12V should be more easily degraded than wild-type, but do not follow through here. They should have a gel comparing complexes of both wild-type and mutant, and also a bar graph quantifying at least one time point with multiple replicates.
3. The section on quantitative assays is very hard to understand. Mechanism 1 should include parameters such as K_s , K_p and k_2 . One presumes that the authors assume rapid binding slow chemistry, though this is not articulated, and likely not justified (see below). What exactly are the parameters from the Kintek Explorer fits and how was k_2/K_s calculated? On the same topic, the fits in Figure S5 appear to go below the points at nitrite = 0, which suggests the equation did not allow for baseline activity. Figure S7 should include the values determined in the global fit and should show the residuals.
4. Figure 6: more information is needed to evaluate this figure, for example, the molecular weights of the proteins. Gel filtration does show that a complex forms and can provide an estimate for the dissociation constant. However, as noted by the authors, it suggests that the on and off rates are slow, which again calls to question the quantitative analysis of RAS cleavage. An SPR experiment might be a better choice here.
5. Figure 8 and associated text: The authors suggest that RAS is present exclusively in the GDP complex in *E. coli*. Only G12V is susceptible to cleavage. Is this consistent with the NMR data? Is the G12V GDP complex more flexible than the wild-type complex? Where is the no imidazole control? What are the molecular weight markers? Is there any independent confirmation that those bands are r1, r2 etc? Does the imidazole-regulated protease work in human cells?

This figure appears to show that RAS cleavage is very selective, yet this is not commented on. Toxicity could be additional evidence of selectivity- is the addition of imidazole toxic to cells expressing G12V?

6. Figure 9: why switch to KRAS? Does the engineered protease also degrade endogenous RAS proteins in tissue culture cells? This should be easy to assess with immunoblotting.

Minor comments (mainly suggestions to make the manuscript more approachable to general readers)

1. The introduction (p3) is confusing to someone with only passing acquaintance to RAS proteins- some description of HRAS and KRAS would be helpful. The sentence describing the steady state populations (line 88-89) should specify that this is the concentration of GTP complexes in cells. Lastly, a brief mention of GAPs and GEFs might help explain the differences observed in bacterial and human cells.

2. Rather than citing classic examples of chemical rescue (p5), the introduction should be more subtilisin focused. The authors should reference the papers where catalytic triad mutants have been rescued with imidazole and nitrite.

3. In the same paragraph, reference(s) are needed for the linkage between S4 and the catalytic triad (line 19- see the Rheinacker et al references of p8, line 187). The final sentence of that paragraph (line 125-128) is misleading because the references demonstrate that subtilisin binds P5-P1', not that cofactor-dependent subtilisins excel in cutting dynamic region.

4. Following paragraph: #1 and #3 are the same.

5. The nmr analysis of the switch 2 region adds context to the protein engineering. However, this focuses on a comparison of GDP and GTP complexes of G12V, rather than a comparison of G12V versus wild-type. Can the authors comment on the flexibility of the switch 2 region in the GDP complexes G12V versus wild-type? Is this complex also more flexible than wild-type?

6. Figure 2A: the flow chart is a nice idea, but could use some more information: how many mutations in SBT149? Label which branch is imidazole dependent versus nitrite dependent, label purpose of each step (engineer S4, etc).

6. Have the authors tested the activity of the engineered proteases against substrates with P4=Ala? Would be nice to include if they have this information.

7. Something was lost between lines 377 and 389. It would help if the authors simply said that endogenous nitrite was sufficient to activate the protease.

8. line 402 "naturally occurring precedent" is obtuse- would be clearer if authors said RAS specific proteases exist.

9. line 439- how do the authors know that the switch region sequence is normally found in amphipathic helices? This information would be useful earlier.

10. The zymogen activation model is interesting, but detracts from the major findings. Perhaps it would be sufficient to simply say that the substrate catalyzes folding and dissociation of the I domain.

11. Figures S2 and S3 are difficult to read. It would be better to use one y-axis, but with a log scale.

Reviewer #2 (Remarks to the Author):

The authors describe a rational design of subtilisin mutants to target active RAS which is involved in cancer. They show by NMR that a region known as switch 2 in RAS is more flexible when the protein is active, therefore being more prone to protease degradation. Several subtilisins were designed to cleave this region and together with the specificity for active RAS, proteases variants were designed to be activated by two different co-factors (imidazole or nitrite). Some in vitro data indicate that the objective is at least partially achieved. The paper provides an elegant and interesting approach to an important problem.

However, I found the paper hard and laborious to be read due to the way it was presented and to the lack of consistency which I will try to point below:

1- Although several subtilisins were designed and produced, experiments were done only to one of them or to a small subset of them depending on the experiment. For example, only SBT2233 was tested in mammalian cells. How about the other proteases? Were they also tested? Why this was not even mentioned? In some experiments, like gel assays for cleavage of RAS(GDP) and RAS(GMPPNP), the statement "data not shown" was used for promising subtilisins variants like SBT2233 and SBT2208. Neither mass spectrometry data is provided for them. In other words, the authors validated different subtilisins than the one used in the cell assay.

2- The gel filtration experiment to determine SBT2208 affinity to the substrate is not adequate. It would be necessary to have controls for the separate proteins loaded in the column. ITC or other binding methods like microscale thermophoresis would be much reliable methods to calculate the affinity. And again, this was shown for only one protease variant.

3- In figure 2A, 4 subtilisins that are triggered by imidazole are indicated. However, co-factor concentration x activity data is provided for only one of them. Specially for SBT2208 that was used in several experiments in presence of imidazole, we do not know the activity in absence of it.

I suggest the authors to rewrite the manuscript in a more organized way. If the authors do not want to show or to perform characterization and/or validation experiments to all the variants, I recommend an explicit table denoting what was done for each of the proteins. This will fortunately make the manuscript more transparent.

Reviewer #3 (Remarks to the Author):

<Major comments>

Comment #1:

The authors used a variety of engineered subtilisin mutants to examine in vitro activity. This led to an interesting observation such as substrate specificity and cofactor selection. This is an important observation. However, since the characteristics of each subtilisin mutants is not summarized in the same table, we cannot easily understand why authors selected individual mutant in each experiment, then we are confused in some case.

Comment #2:

In Figure 1 C, the $\Delta\delta$ is shown both for H-Ras G12V and K-Ras G12V. However, it is not clear how the $\Delta\delta$ for K-Ras was calculated. If the authors measured ^{15}N -HSQC spectra of K-Ras and then the signals were assigned as with H-Ras (line 146-148), the chemical shift table should be deposited in

BMRB. If the NMR data in BMRB has already been deposited, they should clarify the corresponding accession code.

Comment# 3:

The data organization in Fig1D and the corresponding statements in the text are somehow confusing us. In lane 163-164, the authors mention the increased dynamics in Switch 2 of G12V compared to wild type. Is this comment for GDP-bound form or GTP (GMPPNP)-bound form? I guess the conclusion in this section may be the increased Switch 2 flexibility in the GMPPNP-bound form of G12V compared to that of wild type. Therefore, they should add 1H_15N_steady state heteronuclear NOE value data in Fig. 1D.

Comment #4:

In Fig.5 experiment, the authors used H-Ras wild type in complex with GDP and a non-hydrolysable GTP analogue GMPPNP. Since GMPPNP is not a natural substrate of Ras, H-Ras-GMPPNP exhibit slower conformational equilibrium between the two states (I mean it exhibits artificial flexibility), compared to Ras-GTP. Although the corresponding experiment with H-Ras-GTP is difficult because of its intrinsic fast GTP-hydrolysis activity, they should add the results by using natural GTP-bound H-RasG12V, if their goal in this work is to develop a precisely-controlled-Ras specific protease which is effective in cells as emphasized in their discussion. The same is true of Fig. 6.

Comment #5:

In lane 307-315 and Fig.S6, they mention that the cleaved product i.e., 1-67aa ending at YSAM (*) was successfully expressed and purified (in bacterial cells) and subjected to the analysis for inhibitory effect on the protease activity against QEEYAM-AMC substrate. Is this polypeptide* folded or unfolded? Is its conformation (folded or unfolded conformation) essential for the inhibitory effect on the protease activity in the complex? This issue links to the interpretation of Fig. S6 results. Dose the HSQC spectrum of the polypeptide* partially overlap with that of the cleaved H-Ras products? The authors should clarify by NMR that the conformational state (folded or unfolded) of the polypeptide* is substantially consistent with that of the N-terminal cleaved product of H-RAS shown in Fig, S6.

Comment #6:

Fig. 8 and supplemental Figs (S13, S14) show time course of proteolysis of H-Ras wild type-GDP and H-RasG12V-GTP. Is there any evidence to prove that the arrows (in the figures) correspond to each protein? Since many bands are overlapped (crowded) in each lane, isolation of Ras or degradation products is quite difficult. Western immune-detection will be needed at least for H-Ras. Is the protein amount of H-RasG12V(GTP) not changed after proteolysis? Why?

I think r2 (and also r1) fragment of H-RasWT (maybe GDP-bound form) is gradually increased. Is this observation correct? If it's correct, why?

Comment#7:

Figure 9 shows the engineered protease activity in mammalian cells. However, there is no information about cell type in the text, figure legend and Materials and Methods (HEK293T?). Further, the authors transfected eGFP-KRas (expression vector), not eGFP-K-RasG12V. In their culture condition (10%FBS: described in Materials and Methods), stable expression of GTP-bound active form of K-Ras(WT) or its maintenance is theoretically impossible without growth factor stimulation. Although expressed eGFP-K-Ras assumes GDP-bound form, SBT 2233 protease activity is unexpectedly so high. Why? Although the expression level of SBT2233 (whose cofactor is nitrite) is very low in the culture condition without nitrite (lane 6), eGFP-K-Ras cleavage is prominent compared to that with nitrite (+) condition. Is there any reason?

The same is true (i.e., expressed eGFP-K-Ras is GDP-bound form) of Fig. 9B. In addition, SBT2233 may digest endogenous K-Ras, N-Ras, H-Ras (if SBT2233 digests GDP-bound form) because QEEYSAM

sequence in Switch 2 is conserved among these three Ras isoforms. Also, the result may reflect the effect of SBT2233 on the off targets possessing QEEYSAM sequence. In order to show the GTP-form specific activity of SBT2233, they need pull-down assay of eGFP-K-Ras by using Ras-binding domain of Raf, which selectively binds to GTP-bound active Ras. Further, intact cell number should be shown in each time line to preclude off-target activity (such as cell toxicity) of this protease.

Minor comments as follows:

Lane 83-84: guanosine exchange factor → guanine nucleotide exchange factor

Lane 144: malleability → flexibility?

Lane146-147: backbone resonance → backbone NMR resonance or NMR backbone resonance

Lane 149: chemical shift perturbations → chemical shift changes

REVISIONS - Engineering protein-specific proteases: targeting active RAS

General: We appreciate the careful review and detailed comments. Reviewers clearly recognized the significance of the work but found a general lack of clarity in the presentation of the results. Accordingly, we have done the following:

- 1) We have rewritten the paper to focus primarily on protease engineering and kinetic characterization.
- 2) We have eliminated results with the RAS mutant G12V.
- 3) We have focused on the engineering and characterization of just two RAS proteases. We have also changed to a less confusing protease nomenclature.

Responses to specific points:

Reviewer #1 - Major Comments

1. The S4 site appears to have a crucial Cl ion that provides specificity for the P4 Tyr residue. Why do the authors believe this density is Cl ion? There does not appear to be any Cl in the crystallization buffer for SBT2208 (p22)- perhaps it is an unlisted counterion for the Bis-Tris propane? I could not find the buffer conditions for the protease assays- do they include Cl? Does selectivity depend on the presence of Cl? This is a crucial information for understanding the origin of specificity at the P4 site.

Thank you to the reviewer for pointing this out. We inadvertently left out details regarding the YSAM and QEEYSAM peptide soaks from the supplemental methods. SBT crystallizes in a number of conditions and the crystals chosen for peptide soaks were taken from wells crystallized in the presence of NaCl and LiCl. The supplemental methods have been updated accordingly. We have also revised the structure section in the paper itself to better explain that other anions or water can substitute for chloride kinetic reactions.

2. Figure 5. The authors have set up the idea that G12V should be more easily degraded than wild-type, but do not follow through here. They should have a gel comparing complexes of both wild-type and mutant, and also a bar graph quantifying at least one time point with multiple replicates.

Our presentation of G12V results caused confusion in multiple places. We decided it would be best if we removed the G12V data from the manuscript and focused on active vs. inactive RAS.

3. The section on quantitative assays is very hard to understand. Mechanism 1 should include parameters such as K_s , K_p and k_2 . One presumes that the authors assume rapid binding slow chemistry, though this is not articulated, and likely not justified (see below). What exactly are the parameters from the Kintek Explorer fits and how was k_2/K_s calculated? Figure S7 should include the values determined in the global fit and should show the residuals.

We have made all the suggested revisions.

On the same topic, the fits in Figure S5 appear to go below the points at nitrite = 0, which suggests the equation did not allow for baseline activity.

The original fits in Figure S5 were to a simple binding equation. We agree this was confusing and not adequately justified. We have replaced figure 5 with a bar graph (Fig. S3) and eliminated the fitting to avoid confusion.

4. Figure 6: more information is needed to evaluate this figure, for example, the molecular weights of the proteins. Gel filtration does show that a complex forms and can provide an estimate for the dissociation constant. However, as noted by the authors, it suggests that the on and off rates are slow, which again calls to question the quantitative analysis of RAS cleavage.

We have added additional plots to the gel filtration experiment. Binding differences after incubation *without* cofactor are dramatic but also complicated and not directly relevant to the kinetic analysis of specificity in the presence of cofactor. Accordingly we moved these data to the supplement.

5. Figure 8 and associated text: The authors suggest that RAS is present exclusively in the GDP complex in E. coli. Only G12V is susceptible to cleavage. Is this consistent with the NMR data? Is the G12V GDP complex more flexible than the wild-type complex?

REVISIONS - Engineering protein-specific proteases: targeting active RAS

We have eliminated results for G12V for the reasons stated above. The reviewer's conclusion is correct however. From an analysis of $\{^1\text{H}\}$ - ^{15}N steady state heteronuclear NOE data, the Switch 2 region is more flexible in G12V-HRAS(GDP) than in WT-HRAS(GDP) on the ps-ns timescale.

Where is the no imidazole control?

We have added this information to supplement (Fig S16).

What are the molecular weight markers?

Added to Fig 7.

Is there any independent confirmation that those bands are r1, r2 etc?

Yes. We have added this description.

Does the imidazole-regulated protease work in human cells?

Presumably but we have not yet demonstrated this. Programming imidazole-triggered cleavage in mammalian cells (and controlling signaling) is major goal of future work but was beyond the scope of this paper.

This figure appears to show that RAS cleavage is very selective, yet this is not commented on.

Yes it is. We have revised the text to point this out.

Toxicity could be additional evidence of selectivity- is the addition of imidazole toxic to cells expressing G12V?

This is a very astute and important observation. We have in fact observed that imidazole causes growth restriction when certain aggressive RAS protease variants are co-expressed with G12V RAS. We have eliminated G12V data here so that the paper is more focused and readable, however, we will describe this phenomenon in a future paper.

6. Figure 9: why switch to KRAS? Does the engineered protease also degrade endogenous RAS proteins in tissue culture cells? This should be easy to assess with immunoblotting.

We switched to KRAS because it is the major therapeutic target and switch 2 is identical in both isoforms. We are using a validated eGFP reporter provided by the RAS Initiative whose focus in part is developing therapeutics against KRAS-driven cancers. We would argue that assessing endogenous RAS cleavage by immunoblotting is not easy. In a successful experiment, the bands simply disappear leading to a variety of potential interpretations in addition to cleavage and eliminating each potential alternative scenario becomes quite a challenge. We chose a reporter system because a decrease in molecular weight of eGFP-KRAS coinciding with a disappearance of KRAS is convincing proof-of-principle evidence that a prokaryotic extracellular scavenger protease can be modified to cleave a target in mammalian cells. A similar approach has been taken in Antic et al. Nat Comm. (2015) to study a naturally-occurring RAS protease and the authors observed a corresponding decrease in molecular weight of their RAS fusion as well.

Minor comments (mainly suggestions to make the manuscript more approachable to general readers)

1. The introduction (p3) is confusing to someone with only passing acquaintance to RAS proteins- some description of HRAS and KRAS would be helpful. The sentence describing the steady state populations (line 88-89) should specify that this is the concentration of GTP complexes in cells. Lastly, a brief mention of GAPs and GEFs might help explain the differences observed in bacterial and human cells.

We have added description of isoforms to the introduction.

2. Rather than citing classic examples of chemical rescue (p5), the introduction should be more subtilisin focused. The authors should reference the papers where catalytic triad mutants have been rescued with imidazole and nitrite.

REVISIONS - Engineering protein-specific proteases: targeting active RAS

We have made these reference replacements.

3. In the same paragraph, reference(s) are needed for the linkage between S4 and the catalytic triad (line 19- see the Rheinhecker et al references of p8, line 187).

References have been added.

The final sentence of that paragraph (line 125-128) is misleading because the references demonstrate that subtilisin binds P5-P1', not that cofactor-dependent subtilisins excel in cutting dynamic region.

These references have been moved and the sentence revised.

4. Following paragraph: #1 and #3 are the same.

Point #1 has been changed to specify a synthetic peptide substrate.

5. The nmr analysis of the switch 2 region adds context to the protein engineering. However, this focuses on a comparison of GDP and GTP complexes of G12V, rather than a comparison of G12V versus wild-type. Can the authors comment on the flexibility of the switch 2 region in the GDP complexes G12V versus wild-type? Is this complex also more flexible than wild-type?

Please see response to point #5:

6. Figure 2A: the flow chart is a nice idea, but could use some more information: how many mutations in SBT149? Label which branch is imidazole dependent versus nitrite dependent, label purpose of each step (engineer S4, etc).

We have rewritten the paper and changed nomenclature to greatly simplify this presentation. We have added other recommended information to Table 1.

6. Have the authors tested the activity of the engineered proteases against substrates with P4=Ala? Would be nice to include if they have this information.

We have. Activity with P4 = A is extremely low however. We have only presented data for the best non-cognate variations.

7. Something was lost between lines 377 and 389. It would help if the authors simply said that endogenous nitrite was sufficient to activate the protease.

We agree that this was confusing. This version only presents data with added nitrite.

8. line 402 "naturally occurring precedent" is obtuse- would be clearer if authors said RAS specific proteases exist.

Good point. Language has been removed.

9. line 439- how do the authors know that the switch region sequence is normally found in amphipathic helices? This information would be useful earlier.

Amino acids in the motif have high α -helical propensity and the spacing between Y and M matches a helical periodicity. We have added this point to the introduction.

10. The zymogen activation model is interesting, but detracts from the major findings. Perhaps it would be sufficient to simply say that the substrate catalyzes folding and dissociation of the I domain.

This model has been removed.

11. Figures S2 and S3 are difficult to read. It would be better to use one y-axis, but with a log scale.

REVISIONS - Engineering protein-specific proteases: targeting active RAS

Figures revised as suggested.

Reviewer #2 (Remarks to the Author):

1- Although several subtilisins were designed and produced, experiments were done only to one of them or to a small subset of them depending on the experiment. For example, only SBT2233 was tested in mammalian cells. How about the other proteases? Were they also tested? Why this was not even mentioned? In some experiments, like gel assays for cleavage of RAS(GDP) and RAS(GMPPNP), the statement "data not shown" was used for promising subtilisins variants like SBT2233 and SBT2208. Neither mass spectrometry data is provided for them. In other words, the authors validated different subtilisins than the one used in the cell assay.

We have revised the manuscript to focus on only two RAS proteases, one with an imidazole cofactor RASProtease(I) and one with a nitrite cofactor RASProtease(N).

2- The gel filtration experiment to determine SBT2208 affinity to the substrate is not adequate. It would be necessary to have controls for the separate proteins loaded in the column. ITC or other binding methods like microscale thermophoresis would be much reliable methods to calculate the affinity. And again, this was shown for only one protease variant.

We have added additional plots to the gel filtration experiment. Binding differences after incubation *without* cofactor are dramatic but also complicated and not directly relevant to the kinetic analysis of specificity in the presence of cofactor. Accordingly, we moved these data to the supplement.

3- In figure 2A, 4 subtilisins that are triggered by imidazole are indicated. However, co-factor concentration x activity data is provided for only one of them. Specially for SBT2208 that was used in several experiments in presence of imidazole, we do not know the activity in absence of it.

Imidazole-dependence is now shown in Figure 3. Without imidazole, the k_{cat}/K_M is $\sim 10 \text{ M}^{-1}\text{s}^{-1}$.

I suggest the authors to rewrite the manuscript in a more organized way. If the authors do not want to show or to perform characterization and/or validation experiments to all the variants, I recommend an explicit table denoting what was done for each of the proteins. This will fortunately make the manuscript more transparent.

As outlined in the general comment above, we believe this revision presents data in a simpler, more organized, and more readable manner.

Reviewer #3 (Remarks to the Author):

<Major comments>

Comment #1:

The authors used a variety of engineered subtilisin mutants to examine in vitro activity. This led to an interesting observation such as substrate specificity and cofactor selection. This is an important observation. However, since the characteristics of each subtilisin mutants is not summarized in the same table, we cannot easily understand why authors selected individual mutant in each experiment, then we are confused in some case.

As summarized in the general comments, we have written a more focused paper. We have consolidated data for mutagenesis in Table 1 and kinetic data in Table 2.

Comment #2:

In Figure 1 C, the $\Delta\delta$ is shown both for H-Ras G12V and K-Ras G12V. However, it is not clear how the $\Delta\delta$ for K-Ras was calculated. If the authors measured ^{15}N HSQC spectra of K-Ras and then the signals were assigned as with H-Ras (line 146-148), the chemical shift table should be deposited in BMRB. If the NMR data in BMRB has already been deposited, they should clarify the corresponding accession code.

The KRAS data was obtained from the literature (Supplement, references 41, 43). This has been clarified in the caption for Figure S11 in the revised manuscript.

REVISIONS - Engineering protein-specific proteases: targeting active RAS

Comment# 3:

The data organization in Fig1D and the corresponding statements in the text are somehow confusing us. In lane 163-164, the authors mention the increased dynamics in Switch 2 of G12V compared to wild type. Is this comment for GDP-bound form or GTP (GMPPNP)-bound form? I guess the conclusion in this section may be the increased Switch 2 flexibility in the GMPPNP-bound form of G12V compared to that of wild type. Therefore, they should add ¹H-¹⁵N_steady state heteronuclear NOE value data in Fig. 1D.

Our presentation of G12V results caused confusion in multiple places. We decided it would be best if we removed the G12V data from the manuscript and focused on active vs. inactive RAS. The main point is that the Switch 2 region of RAS has increased flexibility in the GMPPNP state relative to the GDP state over a wide range of timescales. The NMR data thus supports the hypothesis that the active (GTP) form of RAS will be more susceptible to proteolytic cleavage, particularly in the Switch 2 region.

Comment #4:

In Fig.5 experiment, the authors used H-Ras wild type in complex with GDP and a non-hydrolysable GTP analogue GMPPNP. Since GMPPNP is not a natural substrate of Ras, H-Ras-GMPPNP exhibit slower conformational equilibrium between the two states (I mean it exhibits artificial flexibility), compared to Ras-GTP. Although the corresponding experiment with H-Ras-GTP is difficult because of its intrinsic fast GTP-hydrolysis activity, they should add the results by using natural GTP-bound H-RasG12V, if their goal in this work is to develop a precisely-controlled-Ras specific protease which is effective in cells as emphasized in their discussion. The same is true of Fig. 6.

The suggestion is logical in principle, but not practicable. GTP hydrolysis is just too fast even with the G12V mutant. The use of GMPPNP is extremely well established in the RAS literature to address the common problem of studying the physical properties of substrate complexes.

Comment #5:

In lane 307-315 and Fig.S6, they mention that the cleaved product i.e., 1-67aa ending at YSAM (*) was successfully expressed and purified (in bacterial cells) and subjected to the analysis for inhibitory effect on the protease activity against QEEYAM-AMC substrate. Is this polypeptide* folded or unfolded? Is its conformation (folded or unfolded conformation) essential for the inhibitory effect on the protease activity in the complex? This issue links to the interpretation of Fig. S6 results. Does the HSQC spectrum of the polypeptide* partially overlap with that of the cleaved H-Ras products? The authors should clarify by NMR that the conformational state (folded or unfolded) of the polypeptide* is substantially consistent with that of the N-terminal cleaved product of H-RAS shown in Fig. S6.

Our NMR analysis indicates that both the N- and C-terminal fragments (r1 and r2) of RAS are disordered, consistent with Fig. S9 in the revised manuscript. This can be determined from the HSQC spectrum because all the peaks corresponding to folded RAS disappear when cleavage occurs.

Comment #6:

Fig. 8 and supplemental Figs (S13, S14) show time course of proteolysis of H-Ras wild type-GDP and H-RasG12V-GTP. Is there any evidence to prove that the arrows (in the figures) correspond to each protein?

Yes. RAS was made with an affinity tag on both termini. Fragments were purified from the extract in co-expression experiments. This is better described in the revised text.

Is the protein amount of H-RasG12V(GTP) not changed after proteolysis? Why?

The amount of G12V RAS does decrease (but we have eliminated data for G12V in the revised manuscript for the reason described above).

I think r2 (and also r1) fragment of H-RasWT (maybe GDP-bound form) is gradually increased. Is this observation correct? If it's correct, why?

All RAS begins in the GTP form after protein synthesis and is hence susceptible to cleavage. It becomes much more resistant after GTP hydrolysis. This is better explained in the revised text.

REVISIONS - Engineering protein-specific proteases: targeting active RAS

Comment#7:

Figure 9 shows the engineered protease activity in mammalian cells. However, there is no information about cell type in the text, figure legend and Materials and Methods (HEK293T?).

HEK 293T are human embryonic kidney cells and widely used for cell-based experiments such as those described in the manuscript. We have added a reference to MATERIALS and METHODS

Further, the authors transfected eGFP-KRas (expression vector), not eGFP-K-RasG12V.

We have removed the G12V sections of this manuscript as those data contributed to the manuscript being unfocused (reviewer 1) and laborious to read (reviewer 2).

In their culture condition (10%FBS: described in Materials and Methods), stable expression of GTP-bound active form of K-Ras(WT) or its maintenance is theoretically impossible without growth factor stimulation.

We did not create a stable cell line as it was unnecessary for these experiments. This was a transient transfection.

Although expressed eGFP-K-Ras assumes GDP-bound form, SBT 2233 protease activity is unexpectedly so high. Why?

We do not have any data quantifying the amount of active versus inactive eGFP-KRAS. It is not correct that eGFP-KRAS assumes the GDP-bound form, however. eGFP-KRAS will localize to the plasma membrane and be subject to the same type of regulation by RAS GAPs and GEFs as the endogenous RAS in these cells. The extent of that regulation is not known, but the reporter used is a validated construct developed by the RAS Initiative.

Although the expression level of SBT2233 (whose cofactor is nitrite) is very low in the culture condition without nitrite (lane 6), eGFP-K-Ras cleavage is prominent compared to that with nitrite (+) condition. Is there any reason?

We suspect this is due to endogenous nitrite and other anions that can activate the protease (e.g. see Gallagher et al. reference #5 in the manuscript). We added nitrite solely to ensure that sufficient nitrite was present to facilitate protease activation. Based on these observations, the experiments are essentially redundant and to improve the focus of the manuscript we have removed the (-) nitrite data and the accompanying text.

The same is true (i.e., expressed eGFP-K-Ras is GDP-bound form) of Fig. 9B. In addition, SBT2233 may digest endogenous K-Ras, N-Ras, H-Ras (if SBT2233 digests GDP-bound form) because QEEYSAM sequence in Switch 2 is conserved among these three Ras isoforms.

It is correct that the protease may digest endogenous HRAS, KRAS, and NRAS. As described above, eGFP-KRAS will be a mixture of active and inactive forms just like the endogenous RAS isoforms.

Also, the result may reflect the effect of SBT2233 on the off targets possessing QEEYSAM sequence. In order to show the GTP-form specific activity of SBT2233, they need pull-down assay of eGFP-K-Ras by using Ras-binding domain of Raf, which selectively binds to GTP-bound active Ras.

Based on this comment, the text appears to imply that we interpret these findings to indicate only active eGFP-KRAS is cleaved by our protease. That is not our interpretation as our other data shows cleavage of inactive RAS, albeit less efficiently. Moreover, such a study is beyond the scope of the present manuscript. We have clarified the text accordingly and agree with the reviewer that as our design efforts progress, determination of e.g. the relative rate of active RAS degradation in mammalian cells for different designs would be informative.

Further, intact cell number should be shown in each time line to preclude off-target activity (such as cell toxicity) of this protease.

We have not found expression of the protease to affect viability of HEK 293T cells in previous experiments. The cells in these experiments showed no signs of viability issues (e.g. sloughing off the tissue culture plate).

REVISIONS - Engineering protein-specific proteases: targeting active RAS

Minor comments as follows:

Lane 83-84: guanosine exchange factor -* guanine nucleotide exchange factor

Lane 144: malleability -* flexibility?

Lane 146-147: backbone resonance -* backbone NMR resonance or NMR backbone resonance

Lane 149: chemical shift perturbations -* chemical shift changes

All changes were made as suggested.

Reviewers' comments:

Reviewer #1 (Remarks to the Author):

The authors have addressed all my comments and the manuscript is much improved. Acceptance is recommended. A few comments/typos are noted below.

1. p4, "and (10,12) 2)" should probably be "(10,12) and 2)"
2. p8, "RASProtease(N)with" should be "RASProtease(N) with"
3. p10, "An additional Interesting" should be "An additional interesting"
4. p11, the authors should be able to estimate half-lives for the data in Figure 5, which would be interesting to know
5. p12, the fonts in Figure 5 are too small, especially in panel A
6. p13, the residuals might be better displayed in separate panel above or below the progress curves
7. p14, Table 2, it might be better to put the units in the column headers
8. p16, "FInally" should be "Finally"
9. Figures S18-S20 regarding protein delivery are not mentioned in the main text. These data could probably be best saved for a future manuscript.
10. p22, the authors imply, but do not explicitly say, that the higher concentrations of nitrite in cancer cells may provide a strategy to selectively cleave RAS in cancer cells. This important point should not be left unsaid.
11. p29, Supplemental Results, Ras cleavage in human cells "we have established that a RAS-specific protease can self-activate..."- these are important points that should be included in the main text

Reviewer #2 (Remarks to the Author):

The authors have made a substantial reorganization on the manuscript. By simplifying protease nomenclatures, providing a table with mutations and withdrawing nonessential assays, the text became clearer, more readable and is now bringing more significance to the main findings.

The authors report the generation of two subtilisin mutants that are specific for the GTP bound form of RAS. The proteases were named RASProtease(N) and RASProtease(I) which are activated by nitrite and imidazole respectively. The authors described assays in synthetic substrates that guided the design of mutations and culminated in the generation of final proteases. The activity of the proteases in the substrate RAS was measured in vitro and in vivo, the latter by using the important therapeutic target KRAS.

Although I recognize the relevance of the work, I have some major concerns about the absence of controls in most of the experiments which I will point out below:

Figure 2 shows the specificity of protease2(I) using synthetic substrates. For a proper comparison it is important to show the data of the progenitor protease. How one can compare that the specificity is changing upon mutations? Also, the data of protease2(N) should be included. The three proteases can be shown in the same figure.

Figure S3 is very important because it shows the dependence of nitrite by Protease2(N) and RASProtease(N). Same experiment should be done for subtilisinBPN', Protease2(I) and RASProtease(I). We don't know if the cofactors are capable of activating more than one protein and more important, the initial progenitor subtilisinBPN'. These controls are crucial for finding out if the

mutants are specifically activated by the co-factors.

Figure 5 is showing RAS cleavage by RASProtease(N) when incubated by 1mM nitrite. Control in absence of nitrite is indispensable but is not shown. Progenitor protease is also needed for comparison with the mutant. RASProtease(I) data is also not shown.

Figure 6: RASProtease(I) kinetic data is incomplete: 6 time courses are shown in panel A (activated RAS) while only three in panel B (inactivated RAS). RASProtease(N) is not shown here but is displayed in Fig S13 with 3 time courses in panel A (activated RAS) and 2 (inactivated RAS) in panel B.

Figure 8: In panel A, controls are lacking. Active protein was not tested in absence of NaNO₂ while inactivated protein was tested only in absence of NaNO₂. How can this be compared? I did not understand what the authors wanted to show.

Unfortunately, is it hard to find an experiment presented in the manuscript which I don't think it needs better control. The authors have a vast amount of data which certainly required a great effort but each assay needs attention in order to achieve the final conclusions.

Reviewer #1 (Remarks to the Author): The authors have addressed all my comments and the manuscript is much improved. Acceptance is recommended. A few comments/typos are noted below.

1. p4, "and (10,12) 2)" should probably be "(10,12) and 2)" Done: **line 105**
2. p8, "RASProtease(N)with" should be "RASProtease(N) with" Done: **line 214**
3. p10, "An additional Interesting" should be "An additional interesting" Done: **line 269**

4. p11, the authors should be able to estimate half-lives for the data in Figure 5, which would be interesting to know.

We moved this figure to the supplement (Fig S10) because it was not central to the quantitative analysis of specificity and reporting half-lives was misleading because the kinetics of cleavage were not simple exponentials.

Supplement line 232.

5. p12, the fonts in Figure 5 are too small, especially in panel A

Figure 5 was moved to supplement and enlarged. (Fig S10)

Supplement line 232.

6. p13, the residuals might be better displayed in separate panel above or below the progress curves
-Done in the revised Figure 6.

7. p14, Table 2, it might be better to put the units in the column headers Done: **line 312**
8. p16, "FInally" should be "Finally" Done: **line 389**

9. Figures S18-S20 regarding protein delivery are not mentioned in the main text. These data could probably be best saved for a future manuscript.

- These data have been removed.

10. p22, the authors imply, but do not explicitly say, that the higher concentrations of nitrite in cancer cells may provide a strategy to selectively cleave RAS in cancer cells. This important point should not be left unsaid.

Added: *"Perhaps the more important point, however, is that the general principles learned from E. coli makes re-programming cultured human cells appear feasible and may provide strategies to selectively cleave RAS in cancer cells.*

lines 508-510

11. p29, Supplemental Results, Ras cleavage in human cells "we have established that a RAS-specific protease can self-activate..."- these are important points that should be included in the main text

Added: *"To date we have only tested one RAS protease in human cells, but we have established that a RAS-specific protease can self-activate in human cells, locate KRAS at the plasma membrane, and cleave it as indicated by the presence of the eGFP fusion product and the precipitous disappearance of KRAS in Fig. 8.*

lines 502-505

Reviewer #2 (Remarks to the Author):

The authors have made a substantial reorganization on the manuscript. By simplifying protease nomenclatures, providing a table with mutations and withdrawing nonessential assays, the text became clearer, more readable and is now bringing more significance to the main findings.

The authors report the generation of two subtilisin mutants that are specific for the GTP bound form of RAS. The proteases were named RASProtease(N) and RASProtease(I) which are activated by nitrite and imidazole respectively. The authors described assays in synthetic substrates that guided the design of mutations and culminated in the generation of final proteases. The activity of the proteases in the substrate RAS was measured in vitro and in vivo, the latter by using the important therapeutic target KRAS.

Although I recognize the relevance of the work, I have some major concerns about the absence of controls in most of the experiments which I will point out below:

Reviewer #2 -1) Figure 2 shows the specificity of protease2(I) using synthetic substrates. For a proper comparison it is important to show the data of the progenitor protease. How one can compare that the specificity is changing upon mutations? Also, the data of protease2(N) should be included. The three proteases can be shown in the same figure.

Fig. 2: k_{cat}/K_M as a function mutation in the P4 pocket **A)** Protease1(N) variants in 1mM nitrite. **B)** Protease1(I) variants in 10mM imidazole. **C)** Comparison of P4 specificity for Protease2(N) in 1mM nitrite and Protease2(I) in 10mM imidazole for the five highest activity sDXKAM-AMC substrates.

1. Data for protease2(N) added to Figure 2. (**Fig. 2C**)
2. Data showing the specificity of progenitor proteases added to Fig. 2. (**Fig. 2A and B, lines 192-195**)
3. Original figure moved to supplement. (**Fig. S4**)

Supplement, lines 190-192

4. Citation of Gron et al. 1992A and added to document the specificity of natural subtilisins. "A thorough sub-site analysis previously performed on the natural subtilisins BPN' and lentus documents their high activity again a broad range of substrates sequences (37, 38)."

lines 188-190:

Reviewer #2 -2) Figure S3 is very important because it shows the dependence of nitrite by Protease2(N) and RASProtease(N). Same experiment should be done for subtilisinBPN', Protease2(I) and RASProtease(I). We don't know if the cofactors are capable of activating more than one protein and more important, the initial progenitor subtilisinBPN'. These controls are crucial for finding out if the mutants are specifically activated by the co-factors.

Fig. 3: k_{cat}/K_M for the target substrate QEEYSAM-AMC as a function of co-factor **A)** Protease2(N) and RASProtease(N) vs. nitrite concentration. **B)** Protease2(I) and RASProtease(I) vs. imidazole concentration. The activity of a progenitor protease (SBT160) with a complete catalytic triad (D32, H64, S221) is shown for comparison.

1. Figure S3 moved to main manuscript. (**Fig. 3A**).
2. Data added for Protease2(I) and RASProtease(I). (**Fig. 3B**).
3. Data for subtilisin BPN' with nitrite or imidazole added. (**Fig. 3A and B**)
lines 196-200
4. The statement was added: "As expected, none of the anions affect the activity of Protease1(I), and none of the imidazole compounds affect the activity of Protease1(N)."
lines 173-174
5. We also added the text: "For reference, the activity of a previously-engineered protease (SBT160) with a complete catalytic triad and preference for P4 = F or Y is also shown. As expected, SBT160 activity is unaffected by nitrite or imidazole (**Fig 3A, B**)."
lines 211-213

Please note that if specificity data was previously reported in the literature, this is stated in the text with the references. If the specificity data is new to this paper, it is reported in a figure.

Fig. S3: k_{cat}/K_M as a function of naturally-occurring anions and imidazole compounds for Protease1(N) and Protease1(I). HMI: 4-hydroxymethylimidazole ($pK_a = 6.45$); IPA: imidazole-4-propionic acid ($pK_a = 6.77$); IAA: imidazole-4-acetic acid ($pK_a = 6.78$). The substrate was sDFKAM-AMC.

6. Specificity of proteases for their cognate co-factor further documented by the addition of **Fig. S3A and B.**

Supplement lines 181-185.

Please note also that co-factor activation is highly specific and that the structural and mechanistic basis for co-factor activation is also firmly established by x-ray crystal structures of D32G and H64G proteases with co-factor bound.

Reviewer #2 - 3) Figure 5 is showing RAS cleavage by RASProtease(N) when incubated by 1mM nitrite. Control in absence of nitrite is indispensable but is not shown. Progenitor protease is also needed for comparison with the mutant. RASProtease(I) data is also not shown.

1. Figure 5 is not a part of the quantitative analysis and was moved to the supplement. (**Fig S10**).
2. We have added this statement to the text "As would be expected from the peptide assays, no cleavage of RAS was observed in the absence of a cognate cofactor with either enzyme."
lines 228-229
3. Also see Figures 7 and S17 for RASProtease(I) for data in *E. coli* cells with and without imidazole.

Figures 7 – lines 374-377; Figure S17

Supplement lines 291-295.

Please note that if an experimental condition produces no measurable effect, this generally is stated in the text. If the effect is measurable, then this is reported in a figure or table.

Reviewer #2 -4) Figure 6: RASProtease(I) kinetic data is incomplete: 6 time courses are shown in panel A (activated RAS) while only three in panel B (inactivated RAS). RASProtease(N) is not shown here but is displayed in Fig S13 with 3 time courses in panel A (activated RAS) and 2 (inactivated RAS) in panel B.

1.

Fig. 6: Kinetics of AMC release from QEEYSAM-AMC by 100nM RASProtease in the presence of RAS. **(A)** RASProtease(I) + RAS(GMPPNP). **(B)** RASProtease(I) + RAS(GDP). **A and B** were measured in the presence of 1 μ M QEEYSAM-AMC and 1 mM imidazole. **(C)** RASProtease(N) + RAS(GMPPNP). **(D)** RASProtease(N) + RAS(GDP). **C and D** were measured in the presence of 1 μ M QEEYSAM-AMC and 1 mM nitrite. Data points are solid circles. Global fit to mechanism 1 are solid lines. Residuals are plotted above each graph.

RASProtease(I) and RASProtease(N) data have been combined in a new Fig 6A-D. An additional trace has been added to Fig 6D. Please note that more traces in part A does not imply traces are missing from the other panels. Experiments were designed to allow for an accurate global fitting to mechanism 1. Concentrations for RAS in different panels were chosen based on the strength of its interactions with a particular protease.

lines 316-322

Reviewer #2 - 5) Figure 8: In panel A, controls are lacking. Active protein was not tested in absence of NaNO₂ while inactivated protein was tested only in absence of NaNO₂. How can this be compared? I did not understand what the authors wanted to show.

Fig 8: RAS-specific protease activity in cells (A) Western blot analysis of cells co-transfected with eGFP-KRAS and active RASProtease1 shows the appearance of a KRAS cleavage product upon induction of the active protease when probed with an anti-GFP antibody following a GFP pull-down. Sodium nitrite was added to the cell culture medium at a final concentration of 1 mM to mitigate potential variability in cellular nitrite concentrations. Appearance of this product coincides with depletion of a RAS-reactive band when probed with an anti-RAS antibody. Appearance of cleaved eGFP-KRAS also coincides with expression of activated protease that has cleaved its inhibitory I-domain. (B) Induction of the active protease in HEK 293T cells at 24 hours after transfection with nitrite supplemented culture medium results in a marked decrease in GFP fluorescence at 48 and 72 hours after transfection compared to the same cells without induction of protease expression.

1. We have revised **Fig. 8** to illustrate only the main points: A RAS-specific protease can self-activate in human cells, locate KRAS at the plasma membrane, and cleave it as indicated by the presence of the eGFP fusion product and the precipitous disappearance of KRAS.

lines 415-425

2. The results with the inactive variant are still included for completeness because it shows the transfection itself does not affect KRAS, but these results have been moved to the supplement to avoid distracting from the main points. (Fig S19).

Fig. S19: RAS-specific protease activity in cells (A) Western blot analysis of cells co-transfected with eGFP-KRAS and inactive RASProtease1 does not result in the appearance of a KRAS cleavage product upon induction of the inactive protease when probed with an anti-GFP antibody following a GFP pull-down. Likewise, the intensity of the RAS-reactive band remains constant when probed with an anti-RAS antibody. The inactive protease fails to cleave its inhibitory I domain as evidenced by the unprocessed protease band at approximately 35 kDa. **(B)** Induction of the inactive protease in HEK 293T cells at 24 hours after transfection results in no change in GFP fluorescence at 48 and 72 hours after transfection compared to the same cells without induction of protease expression.

Supplement lines 315-324

Reviewer 2 comment) Unfortunately, is it hard to find an experiment presented in the manuscript which I don't think it needs better control. The authors have a vast amount of data which certainly required a great effort but each assay needs attention in order to achieve the final conclusions.

We appreciate the reviewer's input on places where control data were not presented and led to confusion.

REVIEWERS' COMMENTS:

Reviewer #2 (Remarks to the Author):

The authors have addressed all my comments by changing the text and including data to main figures. Also, they reorganised some figures from main text to supplementary material and vice versa. I am satisfied with the answers.